# Early-twentieth-century cold bias in ocean surface temperature observations

Sebastian Sippel[1,2 ✉], Elizabeth C. Kent[3], Nicolai Meinshausen[4], Duo Chan[5], Christopher Kadow[6], Raphael Neukom[7,8], Erich M. Fischer[2], Vincent Humphrey[9], Robert Rohde[10], Iris de Vries[2] & Reto Knutti[2]

The observed temperature record, which combines sea surface temperatures with near-surface air temperatures over land, is crucial for understanding climate variability and change[1-4]. However, early records of global mean surface temperature are uncertain owing to changes in measurement technology and practice, partial documentation[5-8], and incomplete spatial coverage[9]. Here we show that existing estimates of ocean temperatures in the early twentieth century (1900–1930) are too cold, based on independent statistical reconstructions of the global mean surface temperature from either ocean or land data. The ocean-based reconstruction is on average about 0.26 °C colder than the land-based one, despite very high agreement in all other periods. The ocean cold anomaly is unforced, and internal variability in climate models cannot explain the observed land–ocean discrepancy. Several lines of evidence based on attribution, timescale analysis, coastal grid cells and palaeoclimate data support the argument of a substantial cold bias in the observed global sea-surface-temperature record in the early twentieth century. Although estimates of global warming since the mid-nineteenth century are not affected, correcting the ocean cold bias would result in a more modest early-twentieth-century warming trend[10], a lower estimate of decadal-scale variability inferred from the instrumental record[3], and better agreement between simulated and observed warming than existing datasets suggest[2].

Global mean surface temperature (GMST) is a crucial indicator of climate change and is essential for guiding climate policies such as the Paris Agreement[11]. It reflects key aspects of Earth's global temperature variability, such as the response to external forcing and large-scale ocean–atmosphere variability[1,12]. Instrumental GMST datasets, which blend sea surface temperature (SST) with surface air temperature over land and sea ice (LSAT)[13], agree broadly on long-term changes and variability[4]. However, assessing the accuracy and consistency of SSTs and LSATs in the early record is challenging, because of (1) observational uncertainties and biases, (2) incomplete coverage, and (3) different physical processes affecting the sea surface and land air temperatures. First, SSTs and LSATs are derived from different measurement techniques and protocols, introducing distinct biases and uncertainties. Early SST records, collected before the Second World War (1939–1945), primarily utilized ship-based bucket measurements. The transition within the early record from wooden to canvas buckets around the late nineteenth century[14], combined with the shifting patterns of shipping routes and shipping fleets, complicates systematic bias adjustments and adds to their uncertainty. Furthermore, essential metadata are often incomplete or missing[6,15]. Engine-room intake measurements replaced buckets over time, and much work has focused on understanding the biases of buckets relative to this more modern measuring technique[16,17].

The LSAT record is, similar to the SST record, subject to potential biases and uncertainties owing to evolving measurement techniques and practices, such as changing sensor exposure[5]. In addition, land-surface changes such as urbanization can strongly affect local measurements and complicate spatial homogenization. Discrepancies exist in the early record between different land datasets[18], and new corrections result in slightly cooler land air temperatures in the late nineteenth century and the early twentieth century[18]. However, at large spatial scales, the LSAT record is considered more reliable because the bias adjustments are smaller and less systematic than for SSTs[5]. Second, both SST and LSAT datasets are affected by incomplete and time-varying spatial coverage, which is particularly sparse in earlier records[19]. Third, SSTs and LSATs are influenced differently by atmospheric and marine processes, air–sea interactions and regional-scale climate variability, and might therefore show large differences that are nonetheless physically consistent[20]. Recent studies have assessed the agreement of coastal SSTs with nearby land air temperatures, identifying a coastal ocean cold anomaly relative to nearby LSATs in the early twentieth century[7,20]. However, several individual years were in fact

[1]Institute for Meteorology, Leipzig University, Leipzig, Germany. [2]Institute for Atmospheric and Climate Science, ETH Zürich, Zurich, Switzerland. [3]National Oceanography Centre, Southampton, UK. [4]Seminar for Statistics, ETH Zürich, Zurich, Switzerland. [5]School of Ocean and Earth Science, University of Southampton, Southampton, UK. [6]Deutsches Klimarechenzentrum GmbH, Hamburg, Germany. [7]WSL Institute for Snow and Avalanche Research SLF, Davos, Switzerland. [8]Climate Change, Extremes and Natural Hazards in Alpine Regions Research Centre, CERC, Davos, Switzerland. [9]Federal Office of Meteorology and Climatology MeteoSwiss, Zurich-Airport, Switzerland. [10]Berkeley Earth, Berkeley, CA, USA. ✉e-mail: sebastian.sippel@uni-leipzig.de

anomalously cold owing to volcanic forcing and natural variability in this period[10,21], and whether these seemingly conflicting conclusions can be reconciled remains unclear.

Here our objective is to evaluate the consistency of the early LSAT and SST record at the global scale. We develop GMST reconstructions using either LSAT or SST records as input to a statistical learning method (ridge regression[22,23]) that is trained on Coupled Model Intercomparison Project Phase 6 (CMIP6) climate model simulations. We construct a ridge regression model for each monthly LSAT and SST coverage mask (from the land-based Climate Research Unit temperature, version 5 dataset (CRUTEM5) and the Met Office Hadley Centre SST dataset, version 4 (HadSST4), respectively[14,24]) from January 1850 to December 2020, which predicts monthly GMST from the incomplete LSAT and SST fields. We then compile the final reconstructions by using each monthly observational value as the predictor for the respective contemporary statistical model, and subsequently average annually. In other words, the reconstruction first uses climate simulations to learn how to reconstruct GMST from sparse and uncertain pseudo-observations, then predicts the actual GMST from the real observations. From this point forward, these reconstructed time series are referred to as $\hat{T}_{CRUTEM5}^{GMST}$ and $\hat{T}_{HadSST4}^{GMST}$, and the set-up is illustrated in Extended Data Fig. 1. Our approach leverages uncertainty and bias estimates[14,24] developed for CRUTEM5 and HadSST4 during training and validation of the statistical models. This approach balances between regions that are highly informative areas for GMST estimation, such as small tropical islands in LSATs (Extended Data Fig. 2), and regions with low measurement uncertainties and biases, to arrive at robust predictions. Our method successfully reconstructs GMST from SST and LSAT datasets separately, achieving lower reconstruction errors over time as data coverage improves and uncertainties decrease. Reconstruction errors are about 15–25% lower than those from traditional methods such as variants of kriging[9] in the early record for both datasets (Methods and Supplementary Fig. 1). Furthermore, we also reconstruct GMST from the unadjusted HadSST4 fields ('HadSST4-unadj'; raw gridded data before any applied corrections; same coverage as HadSST4), a hybrid SST dataset with corrections inferred from coastal weather stations[7] ('CoastalHybridSST'), and night-time marine air temperatures[25] ('ClassNMAT') using the same methodology.

The LSAT-based ($\hat{T}_{CRUTEM5}^{GMST}$) and SST-based ($\hat{T}_{HadSST4}^{GMST}$) reconstructions show high agreement in long-term warming trends, with a reconstructed GMST increase of 1.06 °C (0.92–1.20 °C) and 1.10 °C (1.03–1.17 °C; 2.5th to 97.5th percentile range across the predictions for the CRUTEM5 and HadSST4 ensemble of bias and uncertainty realizations) from 2011 to 2020 compared with 1850 to 1900, for land and ocean records, respectively (Fig. 1). These estimates are similar to the Intergovernmental Panel on Climate Change Sixth Assessment Report reported range[4] of 1.09 °C (0.95 °C to 1.20 °C). The reconstructions agree well on interannual variability, even in periods of sparse coverage (Fig. 1c). From 1850 to 1900, the Pearson correlation (*r*) between annual $\hat{T}_{CRUTEM5}^{GMST}$ and $\hat{T}_{HadSST4}^{GMST}$ is *r* = 0.71, surpassing the raw Pearson correlation (*r* = 0.47) of the global mean time series of CRUTEM5 and HadSST4 over the same period. This correlation strengthens to *r* ≈ 0.80 around the turn of the twentieth century (1875–1925) and to *r* ≥ 0.9 for every subsequent 50-year period starting after 1950. This consistency increases the confidence in the representation of global temperature variability in both land and ocean records despite sparse coverage in the early record.

## A multidecadal ocean cold anomaly

We identify a multidecadal period in the early twentieth century, covering approximately 1900 to 1930, in which $\hat{T}_{HadSST4}^{GMST}$ falls consistently below its land-derived counterpart ($\hat{T}_{CRUTEM5}^{GMST}$; Fig. 1a), on average by about 0.26 °C. This period is the only occurrence of systematic misalignment of the two reconstructions. $\hat{T}_{CRUTEM5}^{GMST}$ indicates steady

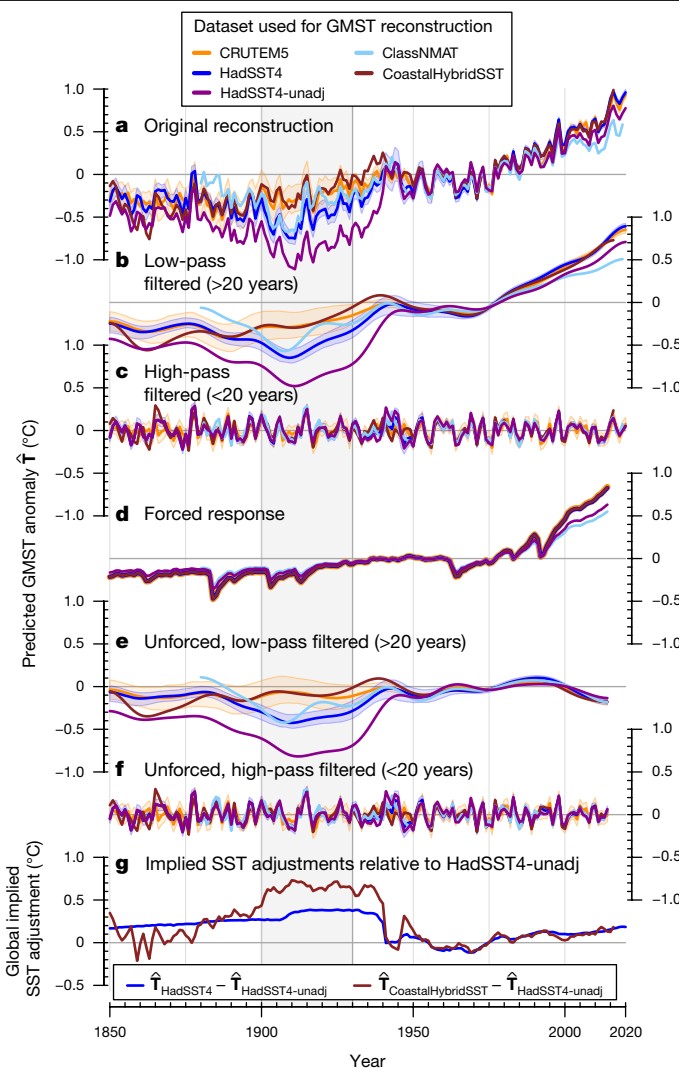

**Fig. 1 | GMST reconstruction from land and ocean records.** GMST reconstruction from the land air temperature record (CRUTEM5[24]) and the SST record (HadSST4[14]; reconstruction denoted as $\hat{T}_{HadSST4}^{GMST}$). The raw, unadjusted HadSST4 dataset (HadSST4-unadj), marine night-time air temperatures (ClassNMAT[25]) and a hybrid SST product with corrections derived from coastal weather stations (CoastalHybridSST[7]) are similarly reconstructed (Methods) and shown for comparison. **a**, Original GMST reconstructions. **b**, Low-pass filtered reconstructions (>20-year timescale). **c**, High-pass filtered reconstructions (<20-year timescale). **d**, Forced GMST response for each reconstruction. **e**, Unforced, low-pass filtered reconstruction. **f**, Unforced, high-pass filtered reconstruction. **g**, Implied global mean adjustments relative to unadjusted HadSST4 data, shown as the difference between the global reconstructions ($\hat{T}_{HadSST4}^{GMST} - \hat{T}_{HadSST4-unadj}^{GMST}$, and $\hat{T}_{CoastalHybridSST}^{GMST} - \hat{T}_{HadSST4-unadj}^{GMST}$). The shading represents the 95th percentile uncertainty ranges of the $\hat{T}_{HadSST4}^{GMST}$ and $\hat{T}_{CRUTEM5}^{GMST}$ reconstructions, obtained by propagating the HadSST4 and CRUTEM5 ensemble of uncertainty realizations. The bold lines show the median across the ensemble. Grey vertical shading illustrates the 1900–1930 period.

warming from the mid-1880s up to around 1940 (Fig. 1b). Conversely, $\hat{T}_{HadSST4}^{GMST}$ shows cooling for about two and a half decades following the early 1880s. According to the SST-based reconstruction, the early 1900s were thus extremely cold, followed by exceptionally fast warming from about 1910 until 1940. The ocean cold anomaly also arises in the SST-based reconstruction when global mean SST or global mean LSAT are used as alternative reconstruction targets (Supplementary Figs. 3 and 4), and when a machine-learning method is used for global

infilling[26] based on the LSAT or SST record only (Extended Data Fig. 4). The ocean-based global infilling GMST reconstruction is on average 0.20 °C colder than the land-based reconstruction, which is very similar to our reconstructions when derived without adding uncertainty and bias estimates at training time (Extended Data Fig. 4a and Supplementary Fig. 2). The ocean cold anomaly is also evident in uncorrected HadSST4 data ($\hat{\mathbf{T}}_{\text{HadSST4-unadj}}^{\text{GMST}}$; Fig. 1), and in other standard SST datasets (centennial in situ observation-based estimates of the variability of SST and marine meteorological variables, version 2 (COBE2-SST) and extended reconstructed SST, version 5 (ERSSTv5); Extended Data Fig. 5). It appears in marine night-time air temperature data ($\hat{\mathbf{T}}_{\text{ClassNMAT}}^{\text{GMST}}$), but ends much earlier, around 1915. Conversely, the corrected coastal hybrid SST product ($\hat{\mathbf{T}}_{\text{CoastalHybridSST}}^{\text{GMST}}$) shows no such anomaly (Fig. 1), in agreement with $\hat{\mathbf{T}}_{\text{CRUTEM5}}^{\text{GMST}}$ and the alternative Berkeley Earth land air temperature dataset[27] (BEST-Land). $\hat{\mathbf{T}}_{\text{BEST-land}}^{\text{GMST}}$ shows a slightly lower baseline than $\hat{\mathbf{T}}_{\text{CRUTEM5}}^{\text{GMST}}$ but no multidecadal cold anomaly (Extended Data Fig. 5).

The implied global SST adjustment in CoastalHybridSST data, that is, the difference to uncorrected data ($\hat{\mathbf{T}}_{\text{CoastalHybridSST}}^{\text{GMST}} - \hat{\mathbf{T}}_{\text{HadSST4-unadj}}^{\text{GMST}}$), increases markedly before 1900 and remains roughly constant at approximately +0.70 °C until 1940, thus reflecting compensation of the ocean cold anomaly (Fig. 1g, brown line). Conversely, standard HadSST4 adjustments are largely stationary up until 1940 (about +0.34 °C during 1900–1940). Some HadSST4 ensemble members feature a small step change in 1906–1910 to account for the transition from well-insulated wooden buckets to poorly insulated canvas buckets[14] but metadata are often missing, making it hard to estimate an appropriate magnitude for that correction[6]. The 1910–1940 period in the SST record is therefore considered as particularly uncertain[5]. A potentially insufficient correction or incorrect timing of this transition, as well as the combination of measurements made by different fleets during that time, may have introduced biases contributing to the identified global ocean cold anomaly.

In the following, we test the possibility that forcing or natural variability is responsible for the ocean cold anomaly. First, we focus on attribution and subtract the forced response from CMIP6 models using a standard attribution method[28] (Fig. 1d–f). The residual unforced and low-pass-filtered time series of $\hat{\mathbf{T}}_{\text{HadSST4}}^{\text{GMST}}$ confirms that the ocean cold anomaly persisted into the 1930s (Fig. 1e). The Krakatoa volcanic eruption[29] led to cooling in the 1880s (Fig. 1d), but it does not account for the anomaly's multidecadal persistence at the ocean surface, as the volcanically forced SST responses in climate models show minimal long-term effects (Supplementary Fig. 3). Second, we compare the observational reconstruction with climate model simulations from the CMIP6 archive, which capture key ocean–atmosphere variability patterns[30]. The models are masked to observational coverage and processed identically to observations, to derive comparable ocean- and land-based GMST reconstructions. The observed difference between ocean- and land-based reconstructions ($\Delta\hat{\mathbf{T}}_{\text{O-L}}^{\text{GMST}} = \hat{\mathbf{T}}_{\text{HadSST4}}^{\text{GMST}} - \hat{\mathbf{T}}_{\text{CRUTEM5}}^{\text{GMST}}$) during the ocean cold anomaly falls well outside internal variability in ocean–land differences based on all 602 historical CMIP6 simulations ($\Delta\hat{\mathbf{T}}_{\text{O-L,CMIP6}}^{\text{GMST}}$; Fig. 2a), in particular at the multidecadal timescale (Fig. 2b). There is no such discrepancy at the interannual timescale (Fig. 2c). Moreover, the Pearson correlation between the observed land- and ocean-based reconstruction falls below that of any CMIP6 model simulation at the start of the ocean cold anomaly, indicating misalignment in the cooling and warming behaviour of the two reconstructions (Fig. 2d). Hence, neither a response to forcing nor internal variability can explain the ocean cold anomaly, unless all state-of-the-art climate models miss a key, unknown process leading to multidecadal decoupling of the ocean and land temperature record.

Multidecadal warming or cooling periods are typically spatially heterogeneous[31], and the early-twentieth-century warming was no exception[8,10]. We analyse the spatial pattern of the ocean cold anomaly

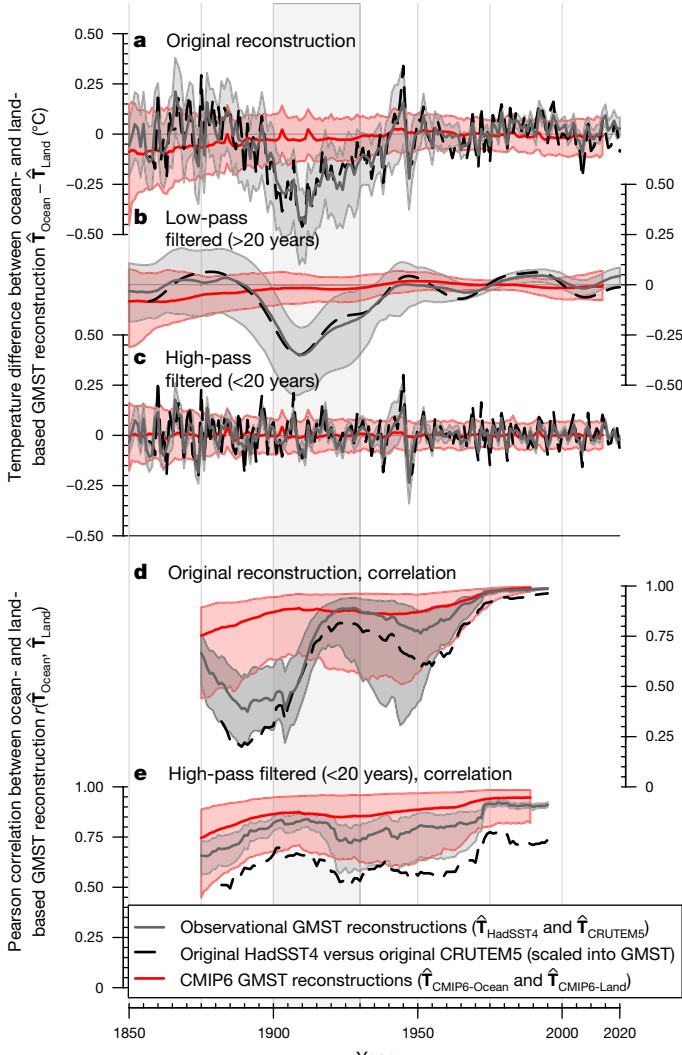

**Fig. 2 | Difference in the GMST reconstructed from ocean and land.** The difference is based on observations (grey and black) and on 602 historical climate model simulations from the CMIP6 archive (red), masked to observed, time-varying coverage. **a**, Temperature difference in observationally derived GMST reconstructions ($\Delta\hat{\mathbf{T}}_{\text{O-L}}^{\text{GMST}} = \hat{\mathbf{T}}_{\text{HadSST4}}^{\text{GMST}} - \hat{\mathbf{T}}_{\text{CRUTEM5}}^{\text{GMST}}$) and the CMIP6 reconstructions. **b,c**, Low-pass filtered $\Delta\hat{\mathbf{T}}_{\text{O-L}}^{\text{GMST}}$ (**b**) and high-pass filtered $\Delta\hat{\mathbf{T}}_{\text{O-L}}^{\text{GMST}}$ (**c**), as well as the CMIP6 reconstructions. **d**, Pearson correlation between ocean- and land-based reconstruction in observations and CMIP6 model simulations in a 51-year moving window. **e**, High-pass filtered Pearson correlation. Black dashed lines represent the original global averages of CRUTEM5 and HadSST4 that are scaled with a land–ocean warming ratio of 1.68 and a global land area and sea ice fraction of 0.33 to represent a comparable 'GMST-like' time series for illustration. Observational reconstruction shading represents 95th percentile uncertainty ranges based on an ensemble of observations with 200 different error realizations, the CMIP6 shading in **a**–**c** is based on the 2.5th to 97.5th percentile across CMIP6 ensemble members, whereas CMIP6 shading in **d** and **e** additionally includes randomly selected error realizations added to the CMIP6 members.

by comparing spatial patterns of average temperature changes between the early twentieth century (1901–1920) and the late nineteenth century (1871–1890). We observe substantial cooling in most ocean regions in HadSST4, including the Atlantic, North Pacific, South Pacific and the Southern Ocean (Fig. 3a). A few sparsely covered regions in the Indian Ocean and central Pacific show slight warming. LSAT, however, does not show sustained and spatially coherent cooling (Fig. 3a). The ocean–land discrepancy is particularly striking when the ocean and land

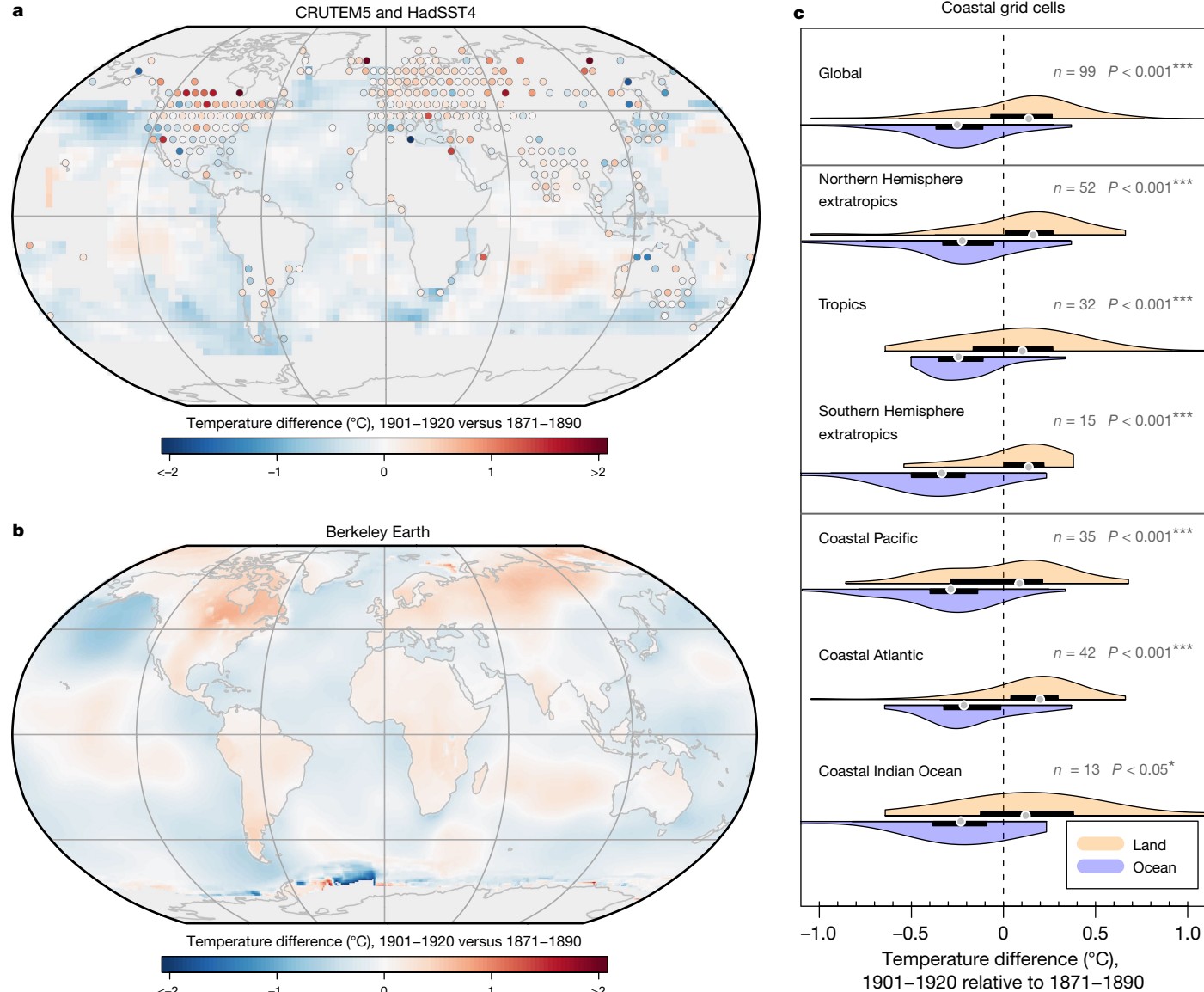

**Fig. 3 | Spatial pattern of turn-of-the-twentieth-century temperature changes.** The difference between post-1900 (1901–1920) and pre-1900 (1871–1890) temperature averages in the SST and LSAT records. **a,b**, LSAT (CRUTEM5, points) and SST (HadSST4, grid cells) in non-infilled datasets (with at least 20% coverage in the 1871–1890 period) (**a**) and in the Berkeley Earth temperature dataset, with land and ocean infilled separately by kriging[27] (**b**). **c**, Individual comparison of SST and LSAT temperature changes at co-located coastal grid cells in different regions in HadSST4 and CRUTEM5. Violins and horizontal bars show the density curve and interquartile range of average temperature changes, respectively. The grey symbols represent the median. Significance is based on *t*-tests between the land and ocean coastal grid cells in each region (*, **, and *** reflects significance at the 5%, 1% and 0.1% level, respectively). Land contours in **a** and **b** from https://www.naturalearthdata.com/.

record are independently spatially infilled such as in the Berkeley Earth dataset, with substantial differences along global coastlines (Fig. 3b). A comparison of coastal grid cells with both land and ocean data shows statistically significant differences in all regions for that period (Fig. 3c), in agreement with previous studies[7,20]. The above indicates that the ocean cold anomaly is manifest in SSTs globally.

## The early twentieth century in palaeoclimate data

To further assess the plausibility of the ocean cold anomaly, we turn to palaeoclimate reconstructions that extend into the twentieth century. The GMST reconstruction from the Past Global Changes 2k working group (PAGES 2k)[12] is based on temperature-sensitive palaeoclimate proxies from land and oceans[32] and seven different statistical reconstruction methods. PAGES 2k GMST shows no substantial temperature changes overall between the early twentieth century (1901–1920) and the late nineteenth century (1871–1890; Fig. 4a). $\hat{\mathbf{T}}_{CRUTEM5}^{GMST}$ falls within the range of these palaeoclimate reconstructions. However, $\hat{\mathbf{T}}_{HadSST4}^{GMST}$ does not overlap with the uncertainty of the palaeoclimate reconstructions. We further analyse marine proxy data from the Ocean2k project reconstructions for three tropical ocean regions[33,34] (Fig. 4b) and individual proxy data. We compare the Ocean2k reconstructions with HadSST4 and LSAT-based regional SST reconstructions. SST estimates suggest strong cooling in all three subregions and the tropics overall from the late nineteenth century to the early twentieth century, whereas Ocean2k and individual palaeoclimate proxies (Extended Data Fig. 6) indicate cooling only in the West Atlantic, with no strong changes in the Indian Ocean and Western Pacific. West Atlantic cooling is supported by the land-based reconstruction of local West Atlantic surface temperatures ($\hat{\mathbf{T}}_{CRUTEM5}^{WAtlantic}$). Overall, the Ocean2k palaeoclimate

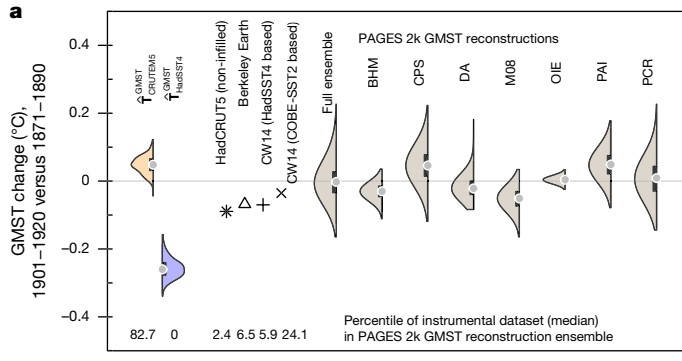

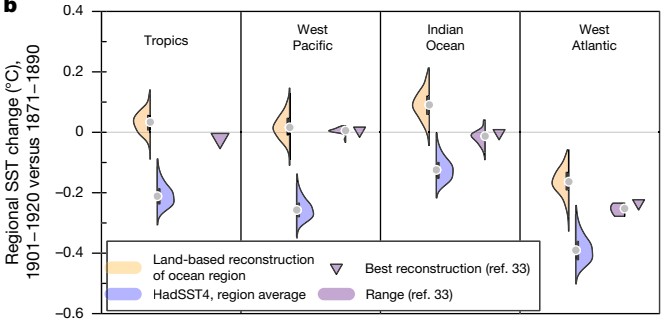

**Fig. 4 | Turn-of-the-twentieth-century temperature changes in palaeoclimate proxy data. a,b,** Difference between post-1900 (1901–1920) and pre-1900 (1871–1890) temperature averages in proxy data from the PAGES 2k GMST reconstructions[12] (**a**) and for proxy data from the Ocean2k reconstruction of tropical ocean temperatures[33] (**b**). The numbers in **a** indicate in which percentile of the PAGES 2k GMST reconstruction ensemble the instrumental datasets fall (medians for $\hat{T}_{CRUTEM5}^{GMST}$ and $\hat{T}_{HadSST4}^{GMST}$). BHM, CPS, DA, M08, OIE, PAI and PCR indicate different palaeoclimate reconstruction methods, explained in Methods. Violins and horizontal bars show the density curve and interquartile range of average temperature changes, respectively. The grey symbols represent the median. CW14, the Cowtan and Way dataset infilled by kriging[19]; HadCRUT5, the Met Office Hadley Centre and Climatic Research Unit global surface temperature data set, version 5[13].

reconstruction aligns more closely with the regional land-based reconstructions than with regional SST estimates; and although Western Atlantic regional cooling into the early twentieth century is evident, a substantial global multidecadal SST cooling trend is not seen in the palaeoclimate data.

## Evidence of residual bias in marine data

The multiple lines of evidence explored above suggest that global gridded SST products contain a cold anomaly during approximately 1900 to 1930 that cannot be explained by known forcings or natural variability. We therefore examine whether it is plausible that currently used SST bias-adjustment methods fail to correctly account for measurement biases in this period. All standard SST products rely on the International Comprehensive Ocean-Atmosphere Data Set (ICOADS)[35], which compiles diverse data sources with varying regional and temporal contributions. During this period, most measurements were taken using various types of bucket[14], expected to be biased cold on average[6]. The methods to adjust HadSST4, ERSSTv5 and COBE-SST2 before the Second World War are based on fixed spatial fields that aim to capture the expected spatial and seasonal variations in measurement offsets associated with particular types of bucket, which are then combined using smoothly varying global weights[15,16]. The period of the ocean cold anomaly featured rapid changes in the contributions of observations made by different national fleets (Extended Data Fig. 7), along with large differences in mean anomalies among

those fleets (Extended Data Fig. 8). Such large and rapidly changing differences between observations from national fleets contributing data in different regions of the ocean cannot be accounted for using the bias-correction methods based on globally constrained and slowly varying adjustment fields, and are hence likely to induce residual biases. Investigation showed that reports from the Historical Sea Surface Temperature Data (HSSTD) project[36], thought to contain mainly US data, is colder than any other data source during 1880 to 1919. Global annual mean anomalies for US data, relative to a modern 1991–2020 climatology[37], are consistently colder than −1 °C; in contrast, UK sources show anomalies closer to −0.5 °C. US observation practices during this period[38] probably induced large cold biases, as observers were instructed to collect samples from three feet below the surface and then immerse the thermometer bulb in the sample on deck for at least 3 minutes before reading, thus exposing the sample to a substantial period of evaporative cooling (Supplementary Information). SST anomalies from the fleets of Germany, the Netherlands and the United Kingdom show an increasing cold offset over the decades 1880–1910, whereas the implied US sources are relatively cold throughout this period (Extended Data Fig. 8). The offset decreases in the subsequent decades, but with slightly different timings. This is consistent with the transition to less well-insulated, but more convenient, canvas buckets over time[5,16], although with the exception of German observations that were made with a different type of bucket. Given the above considerations, we find it plausible that there is a residual cold bias in global SST of a few tenths of a degree that probably causes part of the difference between the land- and ocean-based temperature reconstructions.

## Constraints on early-twentieth-century warming

The implausibility of the ocean cold anomaly around 1900–1930 suggested by our analyses necessitates re-evaluation of key aspects of global temperature variability in instrumental data since 1850: the early-twentieth-century warming from approximately 1900–1940 (or 1900–1950) has been studied widely[10], but the contributing causal factors remain relatively poorly understood. Attribution studies of the observed early warming suggest that about half of this warming is due to external anthropogenic and natural factors[10], implying a significant role for internal multidecadal variability[10,39,40]. However, studies that resolve land and ocean temperatures separately[10,41] show that a much larger fraction of early land temperature changes can be attributed to external anthropogenic and natural forcing, leaving only a smaller role to multidecadal variability. Observed SSTs are cooler than models' SSTs around the turn of the twentieth century and show fast warming thereafter[10,41], thus implying that the large multidecadal internal variability stems primarily from the SST record. In summary, if the ocean cold anomaly is considered to be real, one would conclude that the models underestimate the decoupling between ocean and land warming trends (Fig. 5a), for reasons still unknown. However, multidecadal land warming is in fact tightly coupled to ocean warming according to physical theory[42,43], and for the remainder of the observational record. Here we exploit this relationship and derive observational constraints on ocean warming from our land warming reconstruction ($\hat{T}_{CRUTEM5}^{LSAT}$); and further constraints from coastal hybrid SST data ($\hat{T}_{CoastalHybridSST}^{GMST}$) and the PAGES 2k and Ocean2k palaeoclimate reconstructions[12,33] using the method of emergent constraints[44] (for details, see Methods). Given these constraints, late-nineteenth-century cooling (1871–1910) as depicted by current SST datasets is probably stronger than in reality, and subsequent warming of SSTs during the early-twentieth-century-warming period (1901–1940) is probably too large: the constraints imply a reduced SST cooling (1871–1910) followed by more modest early-twentieth-century-warming in SSTs (1901–1940) (Fig. 5). Very similar results are obtained when 50-year trends are used to derive the constraints (Supplementary Fig. 8; based on 1871–1920

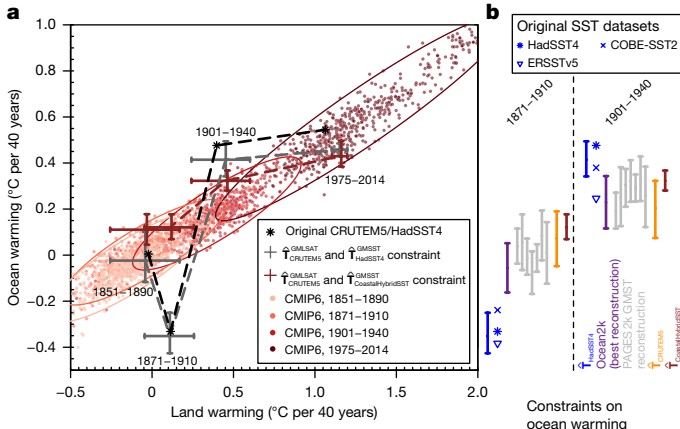

**Fig. 5 | Ocean warming constrained by land warming and palaeoclimate reconstructions. a**, Land and ocean warming on multidecadal timescales is closely linked across CMIP6 models, supported by physical theory[43] (the panel shows original CMIP6 land and ocean warming trends over 40 years for different historical periods). The observed land (CRUTEM5) and ocean (HadSST4) records, and their respective reconstructions, deviate from that relationship owing to the early-twentieth-century ocean cold anomaly. Ellipses show the bivariate 95% range. **b**, Constraints from land air temperature (CRUTEM5), the coastal record (CoastalHybridSST) and palaeoclimate reconstructions (PAGES 2k and Ocean2k) show reduced ocean cooling in the 1871–1910 period owing to a less pronounced early-twentieth-century cold anomaly, followed by more moderate 1901–1940 warming compared with HadSST4 data. Constraints on ocean temperature trends are derived with the method of emergent constraints[44], using the relationships across CMIP6 between the respective temperature trends (GMST from PAGES 2k, $\hat{T}_{CRUTEM5}^{GMLSAT}$ and so on) and global mean ocean temperature trends (see Methods for details). All error bars show 95% prediction intervals. GMLSAT, global mean land surface air temperature; GMSST, global mean sea surface temperature.

and 1901–1950 trends). Both discrepancies are tied to the early-twentieth-century ocean cold anomaly. This is consistent with recent studies that found global temperature variability to be well explained by known anthropogenic and natural forcings, but with the cold anomaly in the early twentieth century remaining a marked residual[2,3]. If this residual is largely owing to uncorrected SST biases, this implies a weaker role of unforced multidecadal variability in the observed record and better agreement with climate models.

## Conclusion

Our GMST reconstructions from land and SST data yield a consistent picture of global interannual temperature variability and long-term changes during the instrumental period, but they highlight a pronounced multidecadal ocean cold anomaly around 1900–1930. Some years in this period were notably cold owing to volcanic activity and internal variability[21]. But such a prolonged cold period does not appear in land data, even with recent homogenization efforts leading to slightly cooler land baseline period temperatures[18], and with new exposure bias-correction methods[45]. The pronounced ocean cold anomaly contradicts the physical theory of land–ocean warming patterns[43], and cannot be explained by internal variability or forcing. The anomaly's global scale corroborates earlier findings of discrepancies between coastal LSAT and SST records[7,20], and palaeoclimate data only suggest regional cooling in the Western Atlantic, rather than a global trend.

In the climate system, understanding often emerges through the integration of different lines of evidence such as physical theory, statistical analysis and historical evidence. This approach identified the Second World War SST bias[46], which was later confirmed by palaeoclimate data[47]. On the basis of the balance of the evidence, we argue that the early-twentieth-century ocean cold anomaly largely stems from uncorrected SST biases, potentially owing to varying source data from national fleets with unaccounted transitions in measurement practice. Continued rescue activities of historical climate data and metadata[48], as well as the development of different SST bias-correction approaches[6], are therefore crucial to better understand this key period, including its regional and seasonal characteristics. Such diverse bias-correction approaches, including group-based SST adjustments[8], and constraints from the SST diurnal cycle[49], from coastal[20] and palaeoclimate data[47], will help ensure that any modifications to the global temperature record are robust. However, the presented constraints based on coastal land temperatures[7,20], statistical analysis and palaeoclimate data imply that agreement between models and reality in the early twentieth century is higher than current observational datasets suggest, and that the role of multidecadal temperature variability is smaller than previously thought.

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

## Methods

### Global mean temperature reconstruction method

To predict global temperature metrics from the LSAT or SST records with incomplete coverage, we invoked a regularized linear regression method known as ridge regression, a key tool in statistical learning[23]. Training of the statistical models was based on climate model simulations from the CMIP6 archive[50] with complete coverage, from which we calculated the target metrics of the statistical model. In the main text, we focus on GMST as the target metric (that is, $\mathbf{Y}^{\text{GMST}}$), but we also reconstructed global mean sea surface temperature (GMSST; Supplementary Fig. 3) and global mean land surface air temperature (GMLSAT; Supplementary Fig. 4). For the land data, we additionally reconstructed regional SSTs of the West Pacific (Fig. 4), Indian Ocean (Fig. 4) and West Atlantic (Fig. 4). The selected target metrics represent key metrics of global temperature change[11] and regional ocean temperature change for the comparison with ocean palaeoclimate reconstructions[33]. Monthly coverage maps of non-infilled data were taken from the CRUTEM5 dataset for the LSAT record[24], and from HadSST4 for the SST record[14]. For each target metric and for both land and ocean, the conceptual reconstruction set-up was conducted using a linear statistical model that relates incomplete spatial LSAT or SST patterns to the respective target metric ($\mathbf{Y}_{\text{mod}}$),

$$\mathbf{Y}_{\text{mod}} = X_{\text{mod}}\,\hat{\boldsymbol{\beta}} + \boldsymbol{\epsilon}. \tag{1}$$

In equation (1), $\mathbf{Y}_{\text{mod}}$ represents a vector of length $n$ (the number of training time steps in the multi-annual monthly time series) of the target metric we seek to reconstruct in climate models used for training. The respective climate models' spatial patterns of LSATs or SSTs are stored in matrices $X_{\text{mod}}$ of dimension $n \times (p+1)$, and $\boldsymbol{\epsilon}$ represents the error terms. Each row contains the spatial pattern for a given training time step ($n$ in total), and each column contains the values of a given location (grid cell) as predictor ($p$ spatial predictors in total, and an additional column of ones that account for the intercept). As SST and LSAT coverage varies over time, the training (equation (1)) is repeated for each monthly coverage mask from January 1850 to December 2020 as the statistical model contains a different number of predictors $p$ for each monthly coverage mask. The training for each monthly coverage mask is based on 93 historical climate model simulations (covering the historical time period 1850–2014, or extended to 2020 where available, with in total $n = 15.626$ model-years, trained for each month separately) from the CMIP6 archive. During the training process, a vector of regression coefficients $\hat{\boldsymbol{\beta}}$ is estimated, representing a weight for each local SST or LSAT time series. Hence, the estimated regression coefficients seek a representation that optimally reconstructs the target metric from the available incomplete spatial pattern of the time series across the climate models used for training. The statistical model and reconstruction set-up is illustrated in Extended Data Fig. 1.

Specifically, for a given coverage mask, here, for example, June 1895, and for GMST as the target metric, the training of our statistical reconstruction model was set up for LSATs and SSTs as:

$$\mathbf{Y}_{\text{mod}}^{\text{GMST}} = X_{\text{mod,Land:1895-06}}\,\hat{\boldsymbol{\beta}}_{\text{Land:1895-06}}^{\text{GMST}} + \boldsymbol{\epsilon}, \tag{2}$$

$$\mathbf{Y}_{\text{mod}}^{\text{GMST}} = X_{\text{mod,Ocean:1895-06}}\,\hat{\boldsymbol{\beta}}_{\text{Ocean:1895-06}}^{\text{GMST}} + \boldsymbol{\epsilon}. \tag{3}$$

It is noted that despite the fact that the coverage is restricted to the June 1895 masks in this example, we still used all months of June in the full time period of 1850–2014 (partly extended to 2020) from the historical simulations of climate models for training the regression models (that is, the estimation of regression coefficients). For example, $X_{\text{mod,Land:1895-06}}$ has the dimensions $n = 15.626$ and $(p+1) = 297$, as the land-coverage mask in CRUTEM5 has $p = 296$ non-missing values in June 1895.

After estimating the regression coefficients, we predicted each target metric at each time step based on SST or LSAT observational datasets, and the actual coverage. For example, for June 1895, LSAT and SST predictions of GMST would read as

$$\hat{\mathbf{T}}_{\text{CRUTEM5:1895-06}}^{\text{GMST}} = X_{\text{CRUTEM5:1895-06}}\,\hat{\boldsymbol{\beta}}_{\text{Land:1895-06}}^{\text{GMST}}, \tag{4}$$

$$\hat{\mathbf{T}}_{\text{HadSST4:1895-06}}^{\text{GMST}} = X_{\text{HadSST4:1895-06}}\,\hat{\boldsymbol{\beta}}_{\text{Ocean:1895-06}}^{\text{GMST}}, \tag{5}$$

where $X_{\text{CRUTEM5:1895-06}}$ is now a $1 \times (p_{\text{Land:1895-06}}+1)$ matrix that contains the actual CRUTEM5 data for June 1895, and $\hat{\boldsymbol{\beta}}_{\text{CRUTEM5:1895-06}}^{\text{GMST}}$ is a vector of length ($p_{\text{Land:1895-06}}+1$, with $p$ reflecting the number of grid cells that contain values in June 1895), which was obtained in the training step (equation (2)). By repeating this step analogously for all time steps from January 1850 to December 2020, we generated an observational reconstruction for each target metric and for the LSAT and SST record, for example, denoted for GMST as $\hat{\mathbf{T}}_{\text{CRUTEM5}}^{\text{GMST}}$ and $\hat{\mathbf{T}}_{\text{HadSST4}}^{\text{GMST}}$. The reconstruction target (GMST) is denoted as the superscript, and the origin of the observational dataset as the subscript. In a similar way, we achieved a reconstruction of each historical CMIP6 model simulation's GMST by projecting masked temperature patterns on the respective regression coefficients (for example, $\hat{\mathbf{T}}_{\text{CMIP6,Land}}^{\text{GMST}}$ and $\hat{\mathbf{T}}_{\text{CMIP6,Ocean}}^{\text{GMST}}$). We averaged the monthly output to annual means. This yielded a CMIP6 global temperature reconstruction that covers 1850–2020 based on only observed coverage, and thus provides a 'like by like' masked comparison with observations (in Fig. 2). The reconstruction as described above in equations (1)–(5) does not account for observational uncertainties and biases in the training process ('no uncertainties and biases in training').

### Including observational biases and uncertainties in GMST estimation

Recent studies have used statistical or machine-learning algorithms, trained on climate model simulations, for optimal infilling of incomplete observations[26]. State-of-the-art climate models indeed represent key modes of ocean–atmosphere variability and forced response patterns[30]. Statistical and machine-learning algorithms thus typically assume that the training distribution (that is, climate models or reanalyses) and the testing distribution (observations) follow the same probability distribution[23]. This is also assumed in the reconstruction method outlined in equations (1)–(5) above. However, the observed temperature record contains substantial measurement uncertainties and systematic biases, in both observed SSTs[6,15] and LSATs[13,24] (summarized in the main text), which climate models do not represent. If those uncertainties and biases were disregarded, our reconstruction algorithm would risk being 'overfitted'—that is, too specifically tailored—to climate model simulated patterns and variability. Hence, it would probably perform rather poorly on observations, even if the relationship between local predictors and GMST was reproduced correctly in the climate model. This phenomenon is known as covariate shift in statistical learning[51]. For example, grid cells in the tropics often carry a high signal for estimating global-scale variability, but precisely those grid cells are sparse in the early observational record, and potentially affected by measurement uncertainties and biases. In addition to the reconstruction method described by equations (1)–(5), we account for measurement uncertainties and biases in our reconstruction by making use of the existing, comprehensive uncertainty and bias models for CRUTEM5 and HadSST4. In HadSST4, uncertainties are modelled in the form of three structurally different components[13,14]: prevalent systematic errors with a complex temporal and spatial correlation structure (biases, denoted $\boldsymbol{\epsilon}_{\text{b,Ocean}}(s, t)$); systematic partially correlated errors from individual ships or buoys ($\boldsymbol{\epsilon}_{\text{p,Ocean}}(s, t)$); and uncorrelated errors

from individual measurements or incomplete sampling of grid-boxes ($\epsilon_{u,Ocean}(s, t)$); indices $s$ and $t$ reflect that the error terms vary in space and time, respectively. The three error terms are treated as statistically independent. The time–space systematic errors are represented by a 200-member ensemble that represents different realizations of these potential biases. Systematic errors from ships or buoys are represented by spatial error covariance matrices, and uncorrelated errors are estimated as gridded error fields.

In CRUTEM5, uncertainties are encoded following the method given in ref. 52. A 200-member ensemble of potential realizations of known, temporally and spatially correlated uncertainties in near-surface air temperature has been produced as part of the HadCRUT5 Noninfilled Data Set[13] (that is, $\epsilon_{b,Land}(s, t)$). The corresponding ensemble of CRUTEM5 LSAT anomalies has been extracted from this, using the HadSST4 ensemble to unblend SST and LSAT anomalies in coastal grid cells. The ensemble realization of biases encompasses uncertainties such as station-based homogenization errors and uncertainty in climatological normals, as well as regional urbanization errors and non-standard sensor enclosures (full description provided in refs. 13,24,52). Uncorrelated uncertainties (measurement errors or incomplete sampling of grid cells) are available in the form of gridded error fields (that is, $\epsilon_{u,Land}(s, t)$).

To take into account these comprehensive error structures, we adjust the matrices that store the appropriately masked model simulated SST or LSAT patterns ($X_{mod}$) for training by randomly adding one systematic error realization ($\epsilon_b(s, t)$) of the HadSST4 or CRUTEM5 ensemble to each CMIP6 historical model simulation used for training. We further generate realizations of the uncorrelated gridded fields ($\epsilon_u(s, t)$), for HadSST4 and CRUTEM5, and in addition a realization of the spatially correlated fields ($\epsilon_{p,Ocean}(s, t)$) for HadSST4. Hence, for a specific coverage mask such as June 1895, we adjust each CMIP6 historical ensemble member in the training sample, obtaining:

$$X^*_{mod,Land:1895\text{-}06} = X_{mod,Land:1895\text{-}06} + \epsilon_{b,Land:1895\text{-}06} + \epsilon_{u,Land:1895\text{-}06}, \quad (6)$$

$$X^*_{mod,Ocean:1895\text{-}06} = X_{mod,Land:1895\text{-}06} + \epsilon_{b,Ocean:1895\text{-}06} \\ + \epsilon_{p,Ocean:1895\text{-}06} + \epsilon_{u,Ocean:1895\text{-}06}. \quad (7)$$

From these perturbed climate model patterns, we train our statistical model as indicated in equations (2) and (3), that is, to optimally predict the simulated (unperturbed) target metric ($Y^{GMST}_{mod}$). This yields a new set of regression coefficients based on the perturbed data ($\hat{\beta}^{*GMST}_{Land:1895\text{-}06}$ and $\hat{\beta}^{*GMST}_{Ocean:1895\text{-}06}$). By adding the error terms, the algorithm optimally predicts the target metric as in equations (1)–(5) while taking into account 'real-world' estimates of uncertainties and biases in the estimation of the regression coefficients. This results in lower coefficient weights for grid cells that are strongly affected by uncertainties and biases (Extended Data Fig. 2). Finally, following equations (4) and (5), we obtain observation-based reconstructions of each target metric ($\hat{T}^{*GMST}_{CRUTEM5}$ and $\hat{T}^{*GMST}_{HadSST4}$), and analogously based on CMIP6 models ($\hat{T}^{*GMST}_{CMIP6,Land}$ and $\hat{T}^{*GMST}_{CMIP6,Ocean}$). An ensemble of 200 observational reconstructed time series ($\hat{T}^{*GMST}_{CRUTEM5}$ and $\hat{T}^{*GMST}_{HadSST4}$) is thus obtained, where the input for each reconstruction contains a different error realization. For ease of notation, we drop the asterisks in the main text, as all observations-based reconstructions are trained with bias and uncertainty realizations added to CMIP6, unless noted otherwise. For CMIP6 reconstructions ($\hat{T}^{*GMST}_{CMIP6,Land}$ and $\hat{T}^{*GMST}_{CMIP6,Ocean}$), individual model members (without error realizations) generally serve as input for GMST reconstructions, with one exception: for the temporal correlations within single reconstructions shown in in Fig. 2d,e, model members are treated as observations, meaning a random error and bias realization is added to each model member. The reconstruction uncertainty ranges in Figs. 1 and 2 are based on the reconstructions ensembles obtained as described above (that is, conditional mean predictions based on our statistical models from 200 different bias and uncertainty realizations).

An illustration of the performance of our statistical reconstruction on CMIP6 models based on sparse coverage (June 1895 for land and ocean, respectively), and more extensive coverage (June 1995 for land and ocean, respectively) is shown as Extended Data Fig. 3. An evaluation of the approach with and without uncertainties and biases is provided in Supplementary Information.

## Statistical learning technique and cross-validation of statistical models

Equation (1) represents a linear regression problem with a relatively high dimensionality, where the number of predictors $p$ can be substantial (for example, $p > 1,000$). Conventional methods such as ordinary least squares aim to minimize the residual sum of squares ($RSS = \sum_i^n (Y_i − X_i\beta)^2$). However, in high-dimensional settings, relying on this single objective may be problematic because regression coefficients lack proper constraints[53]. To address such collinearity issues, we turn to ridge regression, a statistical learning technique designed for such scenarios[23,53]. Ridge regression prevents overfitting by incorporating a penalty for model complexity through the shrinkage of regression coefficients. The shrinkage is based on the sum of squared regression coefficients (referred to as L2 regularization) and a ridge regression parameter $\lambda$ that governs the degree of shrinkage. Consequently, ridge regression addresses a joint minimization problem expressed as

$$\hat{\beta} = \underset{\beta}{\operatorname{argmin}} \left\{ RSS + \lambda \sum_{j=1}^{p} \beta_j^2 \right\}. \quad (8)$$

This results in small yet non-zero regression coefficients, and these coefficients are relatively evenly distributed among correlated predictors[23]. The tuning parameter $\lambda$ determines the extent of shrinkage and is determined through cross-validation, as discussed in the following paragraph. The intercept of the linear model is not shrunk.

## Cross-validation

We used a standard cross-validation approach to determine the ridge regression parameter ($\lambda$) and to obtain the regression coefficients. Cross-validation is a common practice in data science, dividing the raw dataset into different, distinct folds. This ensures that model fitting and validation occur on distinct data subsets, preventing biased performance evaluations. In the context of climate science, the cross-validation method used here adopts a 'leave one model out' strategy, resembling an iterative perfect model approach. In this process, for a total of $k$ CMIP6 models, the ridge regression model is iteratively fitted on $k − 1$ models and validated on the $k$th model (referred to as 'leave-one-model-out cross-validation'). This iterative approach guarantees that the regression coefficients generalize effectively to an unseen model, ensuring the robustness of the statistical model across the CMIP6 multi-model archive. The tuning parameter $\lambda$ is then selected during cross-validation to minimize the mean squared error on out-of-fold data, and the corresponding regression coefficients are extracted. The final set of regression coefficients is obtained as the average across the $k$ model fits. As several climate models provide different numbers of ensemble members, we weight the regression such that each climate model receives equal weight for the extraction of regression coefficients.

## Data pre-processing and observational data

For the gridded fields as regression predictors, we extracted the variables 'tas' (near-surface air temperature) and 'tos' (sea surface temperature) from climate model historical (1850–2014) and 2014–2020 simulations following the Shared Socioeconomic Pathway (SSP) 2-4.5 scenario that contributed to the CMIP6 multi-model archive[50]. All simulations were regridded to a regular 5 × 5° longitude–latitude grid, which is identical to the CRUTEM5 and HadSST4 grid. All climate model data

were centred based on a 1961–1990 reference period, in accordance with CRUTEM5 and HadSST4 data processing.

We computed the target metrics of our regression models as follows from the models' native grids. Global mean surface air temperature (GSAT) is the area-weighted global mean of near-surface air temperature from the spatially complete gridded fields. Global mean surface temperature (GMST) is the blended area-weighted average of SSTs over ocean areas, and surface air temperatures over land areas and areas with sea ice. We follow a standard blending procedure[54], where blending is conducted based on the absolute temperature values (including sea-ice masks from CMIP6 models). GMSST and GMLSAT are determined analogously as the global area-weighted average of SSTs and LSATs, respectively. We also computed regional target metrics for the comparison of instrumental observations with palaeoclimate reconstructions, following ref. 33. The tropical SST reconstruction target metrics cover the (area-averaged) Indian Ocean (20° N–15° S, 40–100° E, in total $25.5 \times 10^6$ km$^2$), western Pacific (25° N–25° S, 110–155° E, $26.9 \times 10^6$ km$^2$) and western Atlantic (15–30° N, 60–90° W, $5.1 \times 10^6$ km$^2$) oceans. All target metrics are centred based on the 1961–1990 reference period.

For the training of our regression models, we selected three historical ensemble members from each CMIP6 model, and the time period 1850–2020 (historical simulations and SSP2-4.5 simulations are concatenated up to 2020). The training dataset consisted in total of 15,626 model-years, and an overview of the CMIP6 models used for training is given in Supplementary Table 4. As the optimal reconstruction coefficients may vary seasonally, training of the regression models for each month $m$ was based on monthly data only from the same month in CMIP6 models (that is, the 15,626 model-years correspond to 15,626 monthly training samples).

Once regression coefficients were extracted following the methodology described above, we obtained observation-based predictions for all target metrics by using the spatially incomplete CRUTEM5 and HadSST4 data as inputs to the regression models (equations (4) and (5)). The comparison of our observation-based reconstruction with CMIP6 models in Fig. 2 was based on 602 CMIP6 historical simulations that contain 'tas' and 'tos' data, that is, 98.835 model-years (overview in Supplementary Table 4), which were masked and reconstructed with observational coverage for each time step.

**Additional observation-based reconstructions**

The focus of this paper is to understand and compare the global temperature reconstructions obtained independently from the HadSST4 dataset[14] and the CRUTEM5 LSAT dataset[24], which are both subject to varying incomplete coverage and affected by complex time-evolving uncertainties and biases. Both datasets and their uncertainties have been developed and maintained over decades[14,24], and both are key datasets that inform the Intergovernmental Panel on Climate Change process[4]. However, several other SST, LSAT and night-time marine air temperature datasets have been developed, and our goal is to compare those data on a 'like by like' basis with our reconstructions: we derived a land-based reconstruction using an alternative LSAT dataset (Berkeley Earth Land[27]) instead of CRUTEM5 as the predictor. These reconstructions ($\hat{\mathbf{T}}_{BEST-land}$) used the model trained with CRUTEM5 coverage, and a few missing grid cells were filled using nearest-neighbour interpolation. Berkeley Earth Land is based on the Global Historical Climatology Network Monthly Temperature Dataset[55], Version 4, with a much higher number of weather stations than CRUTEM5, and compares favourably to a recently homogenized land dataset[18]. Similarly, we used three alternative SST datasets (COBE-SST2[56], ERSST5[57] and an unadjusted version of HadSST4), and we projected all three datasets onto the regression coefficients obtained for the CMIP6 SSTs masked to HadSST4 coverage. COBE-SST2 and ERSST5 in general have larger coverage than HadSST4, but again a few missing grid cells were filled using nearest-neighbour interpolation.

In addition, we trained three global-scale reconstructions based on other datasets for comparison. We followed the reconstruction method outlined above but with different training and observational datasets. First, a reconstruction was trained on CMIP6 marine air temperatures ('tas' over the ocean), and the regression model was subsequently used to derive a reconstruction for observed night-time marine air temperature data (ClassNMAT[25]; that is, reconstructions denoted $\hat{\mathbf{T}}_{ClassNMAT}$ in the main text). Second, we derived a reconstruction based on the CoastalHybridSST dataset[7] ($\hat{\mathbf{T}}_{CoastalHybridSST}$), that is, a SST dataset with corrections derived from co-located coastal weather stations. The training steps were identical to the training of HadSST4, but using the coverage masks from HadSST3 (on which the 'CoastalHybridSST' dataset is based) and its uncertainties. Third, we trained our regression models on CMIP6 data masked to HadSST4 coverage as above, but with the global mean removed individually for each time step. We subsequently used this regression model with HadSST4 with the global mean removed at each time step as predictors (shown in Supplementary Fig. 5). This approach is similar to ref. 58, and allows to test whether large-scale atmosphere–ocean climate variability (which is still largely present if the global mean at each time step is removed) may explain part of the cold anomaly. A full overview of all reconstructions analysed in this paper is provided in Supplementary Table 1, and an overview of the gridded datasets in Supplementary Table 2.

In addition, we compared our observational GMST reconstructions to several widely used blended GMST datasets (Extended Data Fig. 4). In addition, we used the machine-learning method for climate reconstructions by ref. 26 as an independent technique to evaluate and compare the main method of this study (without including estimates of uncertainty and biases at training time) (Extended Data Fig. 4a). Similarly to above, CMIP6 historical experiments were trained with the missing value masks of HadSST4 and CRUTEM5. Subsequently, each observational dataset was infilled by the convolutional neural network using partial convolutions and an updated mask mechanism. An overview over all observational datasets used is provided in Supplementary Table 3.

**Palaeoclimate data**

We compared our reconstructions with two palaeoclimate reconstructions (Supplementary Table 3).

First, we analysed a reconstruction of annual GMST that uses the PAGES 2k temperature multi-proxy data collection[32] (version 2.0.0) as presented in ref. 12. The proxy records undergo screening based on regional temperature, resulting in a subset of 257 records, of which 81% are from trees or corals. The calibration period spans AD 1850–2000, and calibration is based on the Cowtan–Way kriging interpolated HadCRUT4 dataset[19]. Various reconstruction methods are used to reconstruct 2,000 years (AD 1–2000) of annual GMST: composite plus scaling (CPS), principal component regression (PCR), M08[59] (based on a regularized expectation maximization algorithm), pairwise comparison (PAI), the OIE method (derived from CPS but with more comprehensive uncertainty estimation), BHM (a statistical model of GMST depending on external forcing and additive noise), and DA (fusing proxy records and climate model simulations). Four methods (CPS, PCR, BHM and DA) are based on only annually resolved proxies, whereas the other three methods also include low-frequency proxies. Each method generates an ensemble of 1,000 GMST reconstructions for uncertainty quantification; thus, in total, 7,000 ensemble members that cover different proxy types and different statistical reconstruction methods. Method-specific details and adaptations, as well as an evaluation and discussion of uncertainties, are outlined in ref. 12.

Second, we analysed a regional SST reconstruction[33], which is based on only ocean proxy records. The ocean proxy records are derived exclusively from annually or seasonally resolved tropical coral archives. Temperature estimation is based mainly on oxygen isotopic composition ($\delta^{18}O$ of coral carbonate), but records based on the skeletal Sr/Ca

ratio and coral growth rate are also included. Reconstruction targets are regionally averaged tropical SST anomalies at the annual timescale for four large-scale ocean basins: the western Atlantic, the eastern Pacific, the western Pacific and the Indian Ocean. The study uses a weighted CPS approach, using a nesting procedure to address the changing number of available observations over time. Palaeoclimate proxies are not screened, but weighted. Each record's contribution to the composite is scaled by its relationship with instrumental target SST anomalies, considering both magnitude and significance of variance. An ensemble of reconstructions accounts for uncertainties in the CPS method, different weighting schemes and calibration periods. The 'best' reconstruction is selected based on the highest cumulative reduction of error score over the validation period. Finally, we analysed individual palaeoclimate proxies in Extended Data Fig. 6 and Supplementary Information.

## Analysis methods and evaluation metrics

**Timescale separation and attribution.** To analyse our GMST reconstructions, we apply a timescale filtering and an attribution method in Fig. 1. We use a low-pass Butterworth filter with a period of 20 years to separate our original reconstruction in a low-pass filtered and high-pass filtered time series (Fig. 1b,c) based on the R package 'dplr'[60].

To further analyse forced and unforced components of the reconstructions, we apply an attribution method[28] for global mean temperature. The reconstructed $\hat{\mathbf{T}}^{\text{GMST}}$ (or other target metric) is regressed on the multi-model mean of CMIP6 in the 1850–2014 time period (that is, the 'forced response' of the CMIP6 multi-model mean in historical simulations). Internal variability is assumed to follow a simple stochastic AR(1) process, which conceptually represents atmospheric white noise dynamics that force a damped and slower system of the oceans[28]. The regression model is solved following the Hildreth–Lu method[61]. The scaled forced responses for the different reconstructions are shown in Fig. 1d, and unforced residual components in Fig. 1e,f. All data processing and statistical computations as well as figures were created using the R software for statistical computing, version 4.2.2 (ref. 62). For creating the maps in Fig. 3 and Extended Data Figs. 1, 2 and 6, we used the 'sp'[63,64] and 'raster'[65] packages. The land contours and country polygons of the figures that show maps were obtained from Natural Earth (naturalearthdata.com).

## Evaluation metrics and constraints on ocean warming

Several evaluation metrics are derived from our land- and ocean-based reconstructions, and we compare those with climate models and palaeoclimate reconstructions:

- Temperature difference of ocean- versus land-based GMST reconstruction. We analysed the difference between the ocean- and land-based reconstructions at each time step for a given target metric, $\Delta\hat{\mathbf{T}}_{\text{O-L}}^{\text{GMST}} = \hat{\mathbf{T}}_{\text{HadSST4}}^{\text{GMST}} - \hat{\mathbf{T}}_{\text{CRUTEM5}}^{\text{GMST}}$. The observationally derived $\Delta\hat{\mathbf{T}}_{\text{O-L}}^{\text{GMST}}$, including the error realizations, was compared with the equivalently masked reconstruction range of ocean and land temperatures in the CMIP6 models ($\Delta\hat{\mathbf{T}}_{\text{O-L,CMIP6}}^{\text{GMST}}$), and shown in Fig. 2 along with a low-pass and a high-pass filtered version.
- Pearson correlation between ocean- and land-based reconstructions. We calculated the Pearson correlation between the original land and ocean reconstructions ($r(\hat{\mathbf{T}}_{\text{HadSST4}}^{\text{GMST}}, \hat{\mathbf{T}}_{\text{CRUTEM5}}^{\text{GMST}})$) and for the high-pass filtered versions in a 50-year moving window. We compared the time series of $r(\hat{\mathbf{T}}_{\text{HadSST4}}^{\text{GMST}}, \hat{\mathbf{T}}_{\text{CRUTEM5}}^{\text{GMST}})$ with the range of Pearson correlations obtained from reconstructions of CMIP6 model simulations ($r(\hat{\mathbf{T}}_{\text{Ocean,CMIP6}}^{\text{GMST}}, \hat{\mathbf{T}}_{\text{Land,CMIP6}}^{\text{GMST}})$) masked and processed in the same way as observations, including observational error realizations, in Fig. 2.
- Turn-of-the-twentieth-century temperature change (1901–1920 averages compared with 1871–1890 averages). Because ocean-derived temperatures deviate from the land-derived temperatures from the 1890s onwards, we compared the multidecadal period before the ocean cold anomaly (1871–1890) with a period during the cold anomaly (1901–1920). We computed multidecadal

temperature changes around the turn of the twentieth century as the difference between temperature averages in both periods, $\Delta\hat{\mathbf{T}}^{\text{GMST}} = \hat{\mathbf{T}}_{1901-1920} - \hat{\mathbf{T}}_{1871-1890}$. These temperature differences are computed for co-located coastal SST and LSAT data (Fig. 3), and for the global reconstructions and palaeoclimate reconstructions (Fig. 4). A $t$-test for significant differences between co-located coastal land and coastal marine grid cells was conducted for all large-scale regions shown in Fig. 3c.
- Emergent constraints on SST trends. We derived emergent constraints on global mean SST trends in the period 1871–1910 (ocean cooling) and for the early-twentieth-century-warming period (1901–1940). Constraints were derived based on HadSST4 ($\hat{\mathbf{T}}_{\text{HadSST4}}^{\text{GMSST}}$) and the Coastal-HybridSST dataset[7] ($\hat{\mathbf{T}}_{\text{CoastalHybridSST}}^{\text{GMSST}}$), as well as from land temperature change based on CRUTEM5 ($\hat{\mathbf{T}}_{\text{CRUTEM5}}^{\text{GMLSAT}}$), and based on the PAGES 2k GMST reconstructions and the Ocean2k tropical SST reconstruction. We related all datasets to global mean SST trends using an emergent constraint technique[44] explained in the next paragraph.

## Derivation of observational constraints on historical ocean warming

In Fig. 5, we constrained the ranges of global mean SST trends in the 1871–1910 and 1901–1940 periods using the different observation-based datasets (longer periods, 1871–1920 and 1901–1950, are shown additionally in Supplementary Fig. 8). To derive those uncertainty ranges, we applied the method of emergent constraints[44], which uses the relationship between an observable metric (predictor metric) and a target metric across an ensemble of climate model simulations to constrain the target metric based on the observable climate metric. In our application, the predictor metrics and the target metric consisted of temperature trends from the same time period but for different large-scale regions. Our target metric in Fig. 5 was the GMSST trend in the periods 1871–1910 and 1901–1940, respectively. The predictor metrics were the predicted GMSST trends ($\hat{\mathbf{T}}^{\text{GMSST}}$) for the HadSST4 and Coastal-HybridSST datasets, predicted GMLSAT trends ($\hat{\mathbf{T}}^{\text{GMLSAT}}$) for CRUTEM5, GMST ($\hat{\mathbf{T}}^{\text{GMST}}$) for the PAGES 2k reconstructions, and tropical mean SSTs as the weighted average across the tropical Ocean2k regions[33]. We first verified that CMIP6 historical simulations (including individual ensemble members to capture internal variability) show a strong linear relationship between our target metric and the respective predictor metrics. We found that relationships are linear with Pearson correlations of temperature trends of $r(\mathbf{T}^{\text{GMSST}}, \hat{\mathbf{T}}^{\text{GMSST}}) = 0.98$, $r(\mathbf{T}^{\text{GMSST}}, \hat{\mathbf{T}}^{\text{GMLSAT}}) = 0.84$, $r(\mathbf{T}^{\text{GMSST}}, \mathbf{T}^{\text{TMSST}}) = 0.82$, and $r(\mathbf{T}^{\text{GMSST}}, \mathbf{T}^{\text{GMST}}) = 0.96$ in the 1901–1940 period across the CMIP6 historical simulations. The linearity between land and ocean warming is expected from physical theory[43]. Second, we used least-squares linear regression to derive prediction intervals for our target metric based on each individual predictor metric, following ref. 44. Linear regression between two univariate variables $y$ and $x$ is given by $y = \beta_1 x + \beta_0 + \epsilon$, where $\beta_1$ is the regression slope and $\beta_0$ is the intercept. Least-squares linear regression minimizes the least-squares error

$$s^2 = \frac{1}{N-2} \sum_{n=1}^{N} (y_n - f_n)^2, \qquad (9)$$

where $N$ is the number of available samples. The minimization yields an estimate of the regression slope, $\hat{\beta}_1 = \frac{\sigma_{xy}^2}{\sigma_x^2}$, where $\sigma_{xy}^2$ and $\sigma_x^2$ are the covariance between $x$ and $y$, and the variance of $x$, respectively. The 'prediction error' of linear regression is given by[44]:

$$\sigma_f(x) = s\sqrt{1 + \frac{1}{N} + \frac{(x - \bar{x})^2}{N\sigma_x^2}}. \qquad (10)$$

Hence, for an observations-based temperature trend ($x_{\text{obs}}$), we predict the mean conditional GMSST trend as $\hat{y}_{\text{obs}} = \beta_1 x_{\text{obs}} + \beta_0$, and we

obtain the 2.5th to 97.5th percentile prediction ranges as $\hat{y}_{obs} \pm 1.96\sigma_f$. In cases where observational uncertainty estimates are available as an ensemble ($x_{obs,i}$, we derived the mean conditional GMSST trend as above for each member $i$ of the observational ensemble (with $n_{ens}$ denoting the ensemble size), and subsequently we sample for each member $k = 100$ times from a Gaussian distribution ($\mathcal{N}(\mu = \hat{y}_{obs,i}, \sigma^2 = \sigma_f^2)$) to capture the prediction uncertainty around each ensemble member $\hat{y}_{obs,i}$. Finally, we obtain the constrained range of GMSST trends as the empirical 2.5th and 97.5th percentiles across the $n_{ens} \times k$ samples. The obtained prediction ranges thus account for internal variability in the relationship between our respective predictor metrics and our target metric.

### Implications of early-twentieth-century global ocean cold anomaly for unforced variability estimates

In the main text, we discuss the implications of the ocean cold anomaly for the early-twentieth-century land and ocean warming, including the apparent ocean cooling before and up to around 1910 that is not supported by the land data. An additional aspect that requires attention in future work is the representation of decadal-to-multidecadal variability in models and observations. Some long-standing concerns have been raised that climate models may underestimate the magnitude of multidecadal variability at global or subglobal scales[66–68], whereas others argue that global-scale temperature variability on interannual to centennial timescales is plausibly represented[12,69–73]. Yet, the range of unforced interdecadal variability in CMIP6 models is large[74]. Recent studies have found that global temperature variability is well explained by known anthropogenic and natural forcings, but the cold anomaly in the early twentieth century remains a marked cold residual[2,3]. If the ocean cold anomaly arises partly owing to uncorrected SST biases, it would imply improved agreement between models and observations in the instrumental period, and thus a smaller role for internal unforced variability than previously thought owing to a reduced cold residual in the early twentieth century[2,3].

### Data availability

All data used in this study are from publicly available sources. Climate model simulations from the CMIP6 archive are available at https://esgf-node.llnl.gov/projects/cmip6/; the individual models used for training and analysis are detailed in Supplementary Table 4. Gridded observational temperature datasets are available from different sources (detailed overview available in Supplementary Table 2) and summarized here: CRUTEM5[24], HadSST4[14] (including HadSST4-unadj) and HadCRUT5[13] are available from the UK Met Office under British Crown Copyright (https://www.metoffice.gov.uk/hadobs/crutem5/data/CRUTEM.5.0.1.0/download.html; https://www.metoffice.gov.uk/hadobs/hadsst4/data/download.html; https://www.metoffice.gov.uk/hadobs/hadcrut5/), provided under an Open Government License, including its bias realizations, uncertainty estimates, land fraction files; and its respective original global mean temperature estimates. Gridded night-time marine air temperature data from ClassNMAT[25] are available through https://catalogue.ceda.ac.uk/uuid/5bbf48b128bd488dbb10a56111feb36a, and CoastalHybridSST data[7] can be obtained from https://www-users.york.ac.uk/~kdc3/papers/evaluating2017/. Additional gridded SST datasets (COBE-SST2 from ref. 56 and ERSSTv5 from ref. 57, both provided by the NOAA PSL, Boulder, Colorado, USA, from their website at https://psl.noaa.gov/data/gridded/data.cobe2.html and https://psl.noaa.gov/data/gridded/data.noaa.ersst.v5.html) and land air temperature datasets (Berkeley Earth Land from ref. 27, https://berkeleyearth.org/data/) are available. Standard GMST time series shown in Extended Data Fig. 4 are available from the following sources: the Cowtan–Way14 datasets[19] (https://www-users.york.ac.uk/~kdc3/papers/coverage2013/series.html); JMA GMST[56] (https://ds.data.jma.go.jp/tcc/tcc/products/gwp/temp/ann_wld.html);

NOAAGlobalTemp Version 5[75] (https://www.ncei.noaa.gov/products/land-based-station/noaa-global-temp); NASA-GISTEMP[76] (https://data.giss.nasa.gov/gistemp/). The in situ SST analysis used data from the ICOADS database Release 3[35] (https://icoads.noaa.gov/), and from the World Meteorological Organization (WMO) Historical Sea Surface Temperature Data project[36] (HSSTD). The climatological baseline to process in situ records is taken from the European Space Agency Climate Change Initiative (ESA-CCI) SST climate data record (Level 4 SST CCI analysis product[37], available from https://climate.esa.int/en/projects/sea-surface-temperature/. The analysis of palaeoclimate data made use of the PAGES 2k multi-proxy GMST reconstructions[12,32] (https://www.ncei.noaa.gov/access/paleo-search/study/26872), and of the Ocean2k regional sea surface temperature reconstructions in the tropics[33] (https://www.ncei.noaa.gov/access/paleo-search/study/17955), including the tropical average reconstruction[34]. Intermediate and post-processed data to reproduce our analysis are available under https://data.iac.ethz.ch/Sippel_et_al_2024_ocean-cold-anomaly/. Final figures and the reconstructions developed in this study are made publicly and permanently available on Zenodo at https://doi.org/10.5281/zenodo.13646027 (ref. 77). Source data are provided with this paper.

### Code availability

All code to reproduce this analysis is publicly available on Zenodo at https://doi.org/10.5281/zenodo.13646027 (ref. 77).

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

**Acknowledgements** We thank K. Haustein for discussions and comments on an earlier draft of the paper; U. Beyerle, R. Lorenz and L. Brunner for the preparation and maintenance of the CMIP6 data; and F. Loer for help with code checking and documentation. We thank all observers, creators, maintainers and providers of datasets; references, acknowledgements and download links to all datasets are provided in 'Data availability'. We acknowledge the World Climate Research Programme's Working Group on Coupled Modelling, which is responsible for CMIP, and we thank the climate modelling groups for producing and making available the model output. For CMIP, the US Department of Energy's Program for Climate Model Diagnosis and Intercomparison provides coordinating support and led development of software infrastructure in partnership with the Global Organization for Earth System Science Portals. S.S. acknowledges the projects 'Constraints on near-term warming projections via distributionally robust statistical and machine learning' (COPE; grant agreement C22-02, funded by the Swiss Data Science Center), 'Artificial Intelligence for Enhanced Representation of Processes and Extremes in Earth System Models' (AI4PEX; grant agreement 101137682, funded by the EU's Horizon Europe programme), and the climXtreme project (Phase 2, project PATTETA, grant number 01LP2323C, funded by the German Federal Ministry of Education and Research). E.M.F. and S.S. acknowledge funding from the EU Horizon 2020 Project XAIDA (grant agreement 101003469). E.C.K. was supported by UKRI NERC grant NE/S015647/2.

**Author contributions** S.S. identified the research gap and developed the study's core idea supported by R.K. S.S. designed and implemented the main statistical methodology supported by N.M. S.S. analysed all the data, supported by further analysis of palaeoclimate data (R.N.), the SST source data that contribute to ICOADS (E.C.K.), and a machine-learning tool trained to infill CRUTEM5 and HadSST4 (C.K.). S.S. wrote the initial draft and all authors contributed to discussing the results and writing of the final paper.

**Funding** Open access funding provided by ETH Zürich.

**Competing interests** The authors declare no competing interests.

**Additional information**
**Correspondence and requests for materials** should be addressed to Sebastian Sippel.

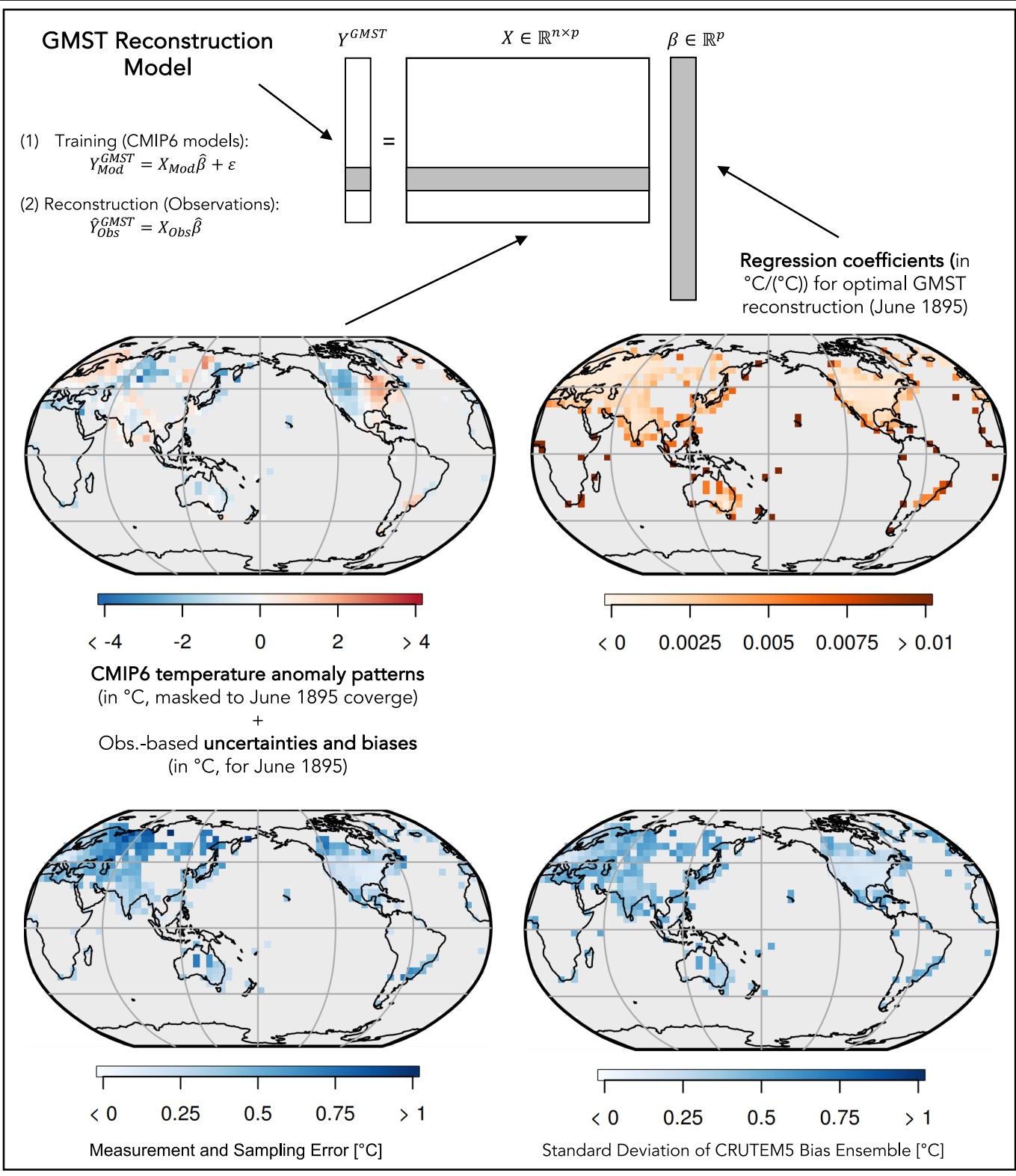

**Extended Data Fig. 1 | Illustration of global mean surface temperature reconstruction.** A regularized linear statistical model relates spatially incomplete patterns of land- or ocean-based temperature measurements in CMIP6 models ($X^{n \times p}$, not including the intercept in the illustration) to the corresponding GMST ($Y^{GMST}$). Uncertainty and bias estimates of the observational data are included in the training process by adding realizations of biases and uncertainties (bottom row) to the masked CMIP6 predictors. CMIP6 simulated patterns and uncertainty and bias estimates are illustrated here for the land coverage in June 1895.

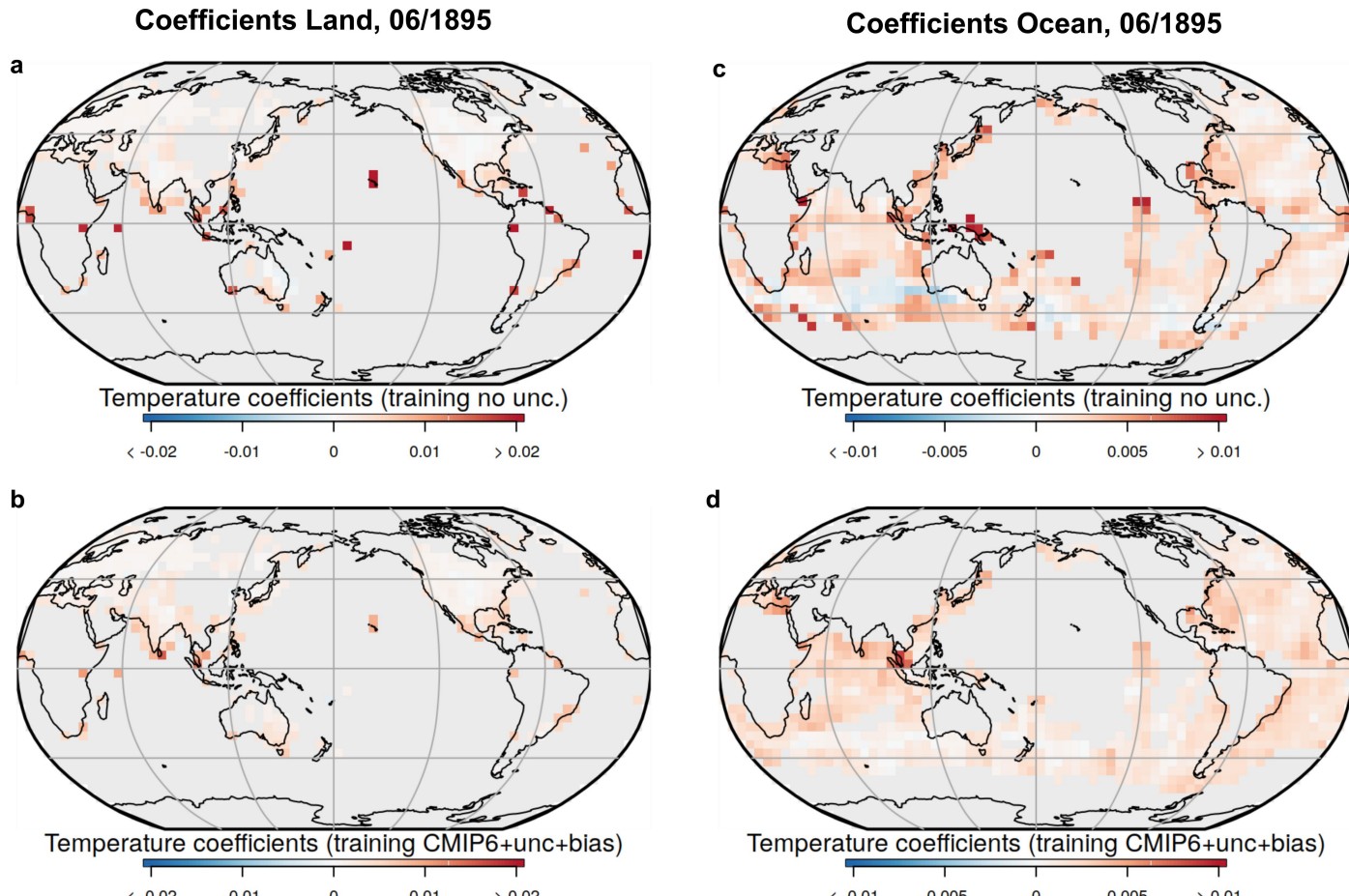

**Extended Data Fig. 2 | Illustration of regression coefficients for land- and ocean-based reconstruction.** Ridge regression coefficients for the GMST reconstruction based on (**a**, **b**) land, and (**c**, **d**) ocean data. Regression coefficients are shown for (**a**, **c**) training on CMIP6 data without adding error realizations, and (**b**, **d**) when error realizations are added.

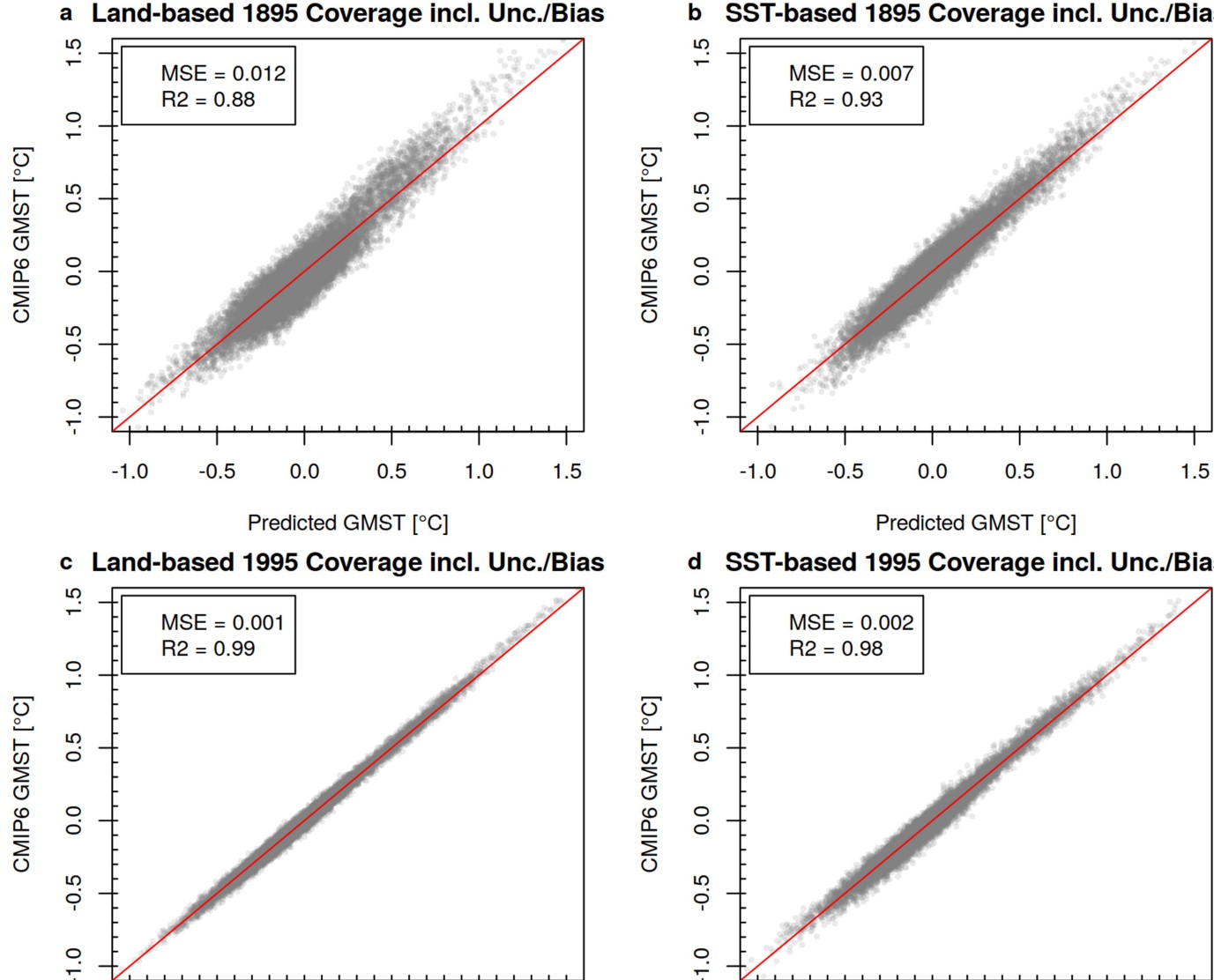

**Extended Data Fig. 3 | Illustration of reconstruction skill for June 1895 and June 1995 coverage; including estimates of uncertainty and biases in the training step.** Each gray dot represents a year of the CMIP6 realizations used for training the reconstructions (1850-2020; coverage fixed to (**a, b**) June 1895 and (**c, d**) June 1995, respectively), with a specific randomized representation of the observational errors, and one fixed bias realization added to each CMIP6 historical simulation.

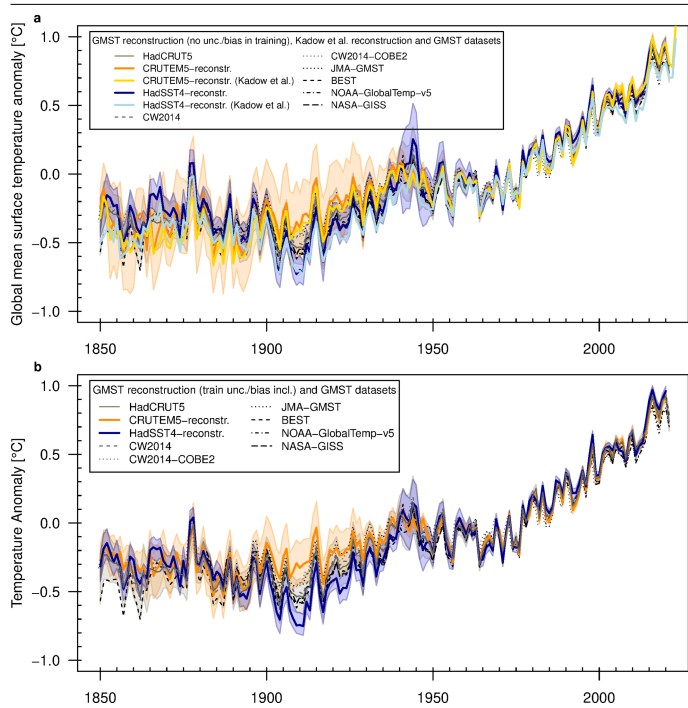

**Extended Data Fig. 4 | Comparison of GMST reconstructions to standard GMST datasets. a** GMST reconstructions when uncertainties are not included in the training dataset, compared to standard GMST products, and **b** GMST reconstructions with uncertainties included in the training dataset. Panel **a** also shows a reconstruction of GMST based on infilling the whole field from land (yellow line) and ocean-data only (light blue line) based on the machine learning method used in ref. 26.

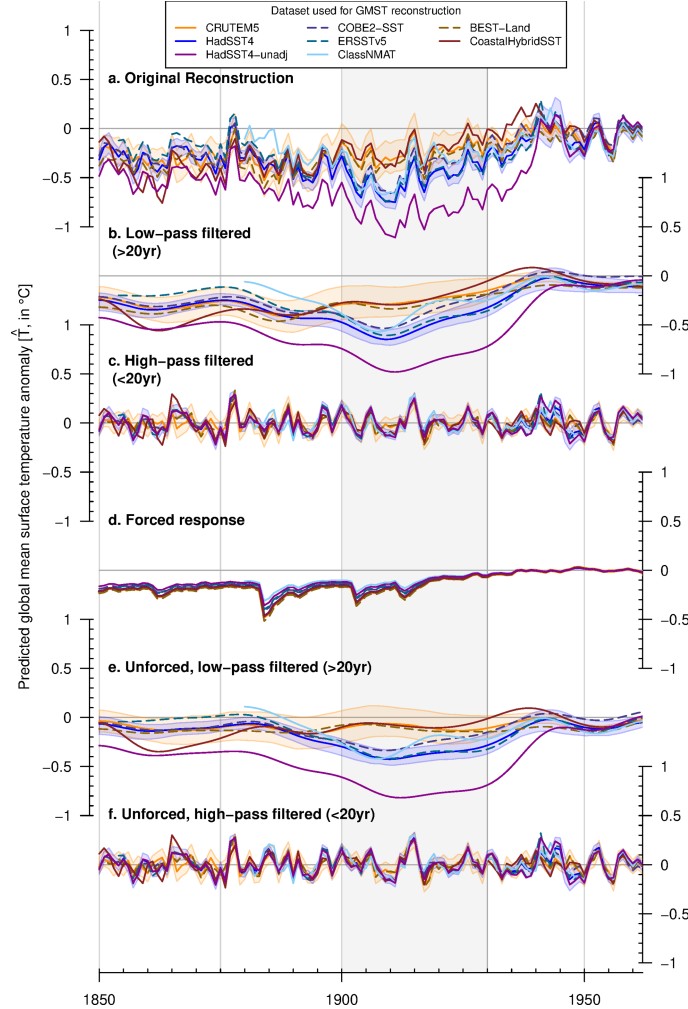

**Extended Data Fig. 5 | Global mean surface temperature reconstruction from the land and ocean record with several additional reconstructions, showing the pre-1960 record.** GMST reconstruction from the SST record ($\hat{\mathbf{T}}^{GMST}_{HadSST4}$) and from the land air temperature record ($\hat{\mathbf{T}}^{GMST}_{CRUTEMS}$), shown along several additional GMST reconstructions from alternative land air temperature data ($\hat{\mathbf{T}}^{GMST}_{BEST-Land}$) and from other SST products ($\hat{\mathbf{T}}^{GMST}_{COBE-SST2}$ and $\hat{\mathbf{T}}^{GMST}_{ERSSTv5}$). **b**. low-pass filtered reconstructions (>20-year time scale), **c**. high-pass filtered reconstructions (<20-year time scale), **d**. forced GMST response for each reconstruction, **e**. unforced, low-pass filtered reconstruction, **f**. unforced, high-pass filtered reconstruction. The additional GMST reconstructions are based on CRUTEM5 and HadSST4 coverage masks, respectively, where few missing grid cells have been filled using the nearest-neighbour technique.

Shading represents the 95th percentile uncertainty ranges of the $\hat{\mathbf{T}}^{GMST}_{HadSST4}$ and $\hat{\mathbf{T}}^{GMST}_{CRUTEMS}$ reconstructions, obtained by propagating the HadSST4 and CRUTEM5 ensemble of uncertainty realizations; bold lines show the median across the ensemble.

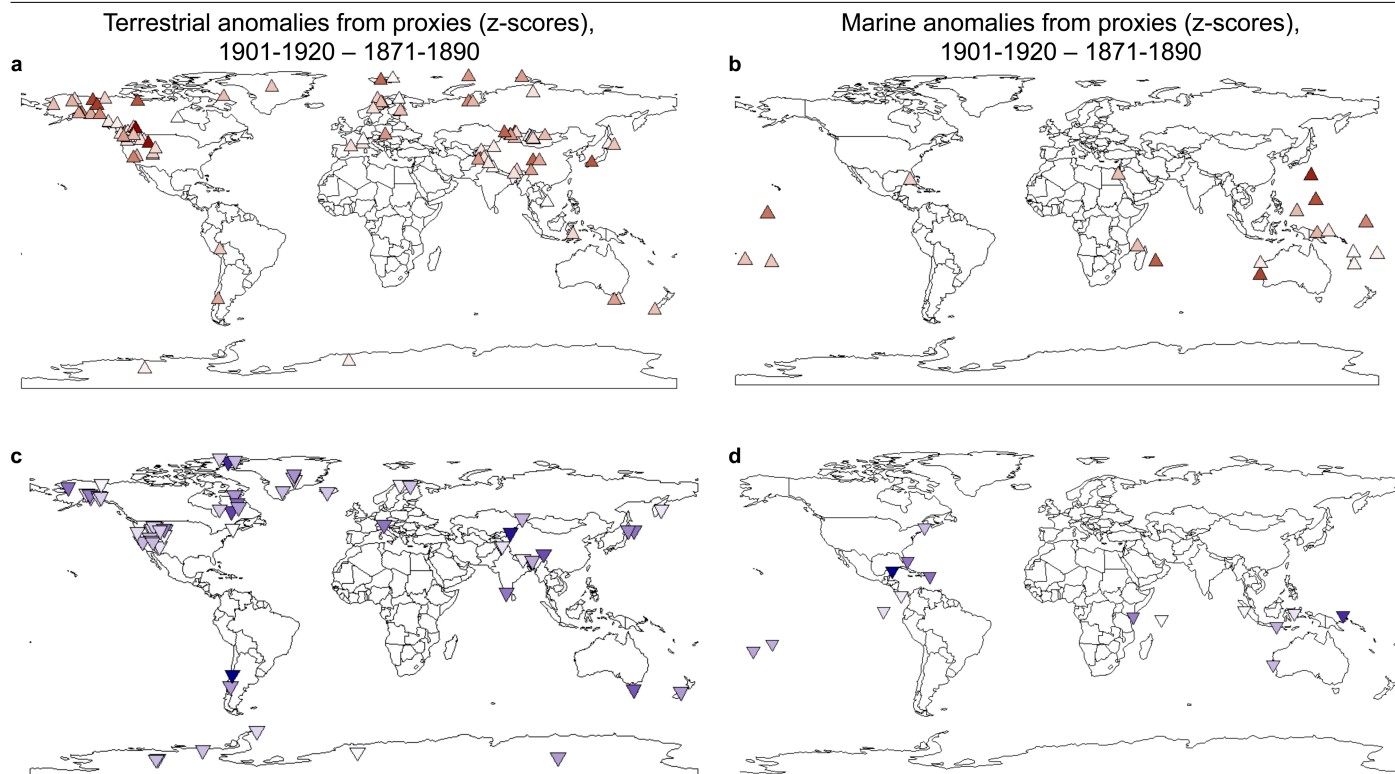

Terrestrial anomalies from proxies (z-scores), 1901-1920 − 1871-1890

Marine anomalies from proxies (z-scores), 1901-1920 − 1871-1890

**Extended Data Fig. 6 | Relative changes in terrestrial and marine temperature proxies in the period 1901-20 compared to 1871-1890.** **a** Terrestrial proxies with positive anomalies, **b** marine proxies with positive anomalies, **c** terrestrial proxies with negative anomalies, **d** marine proxies with negative anomalies. All proxy records are uncalibrated, and are shown as standardized z-scores relative to the 1871-90 reference period. Hence, colours reflect relative warming/cooling; the darker the color, the greater the warming/cooling.

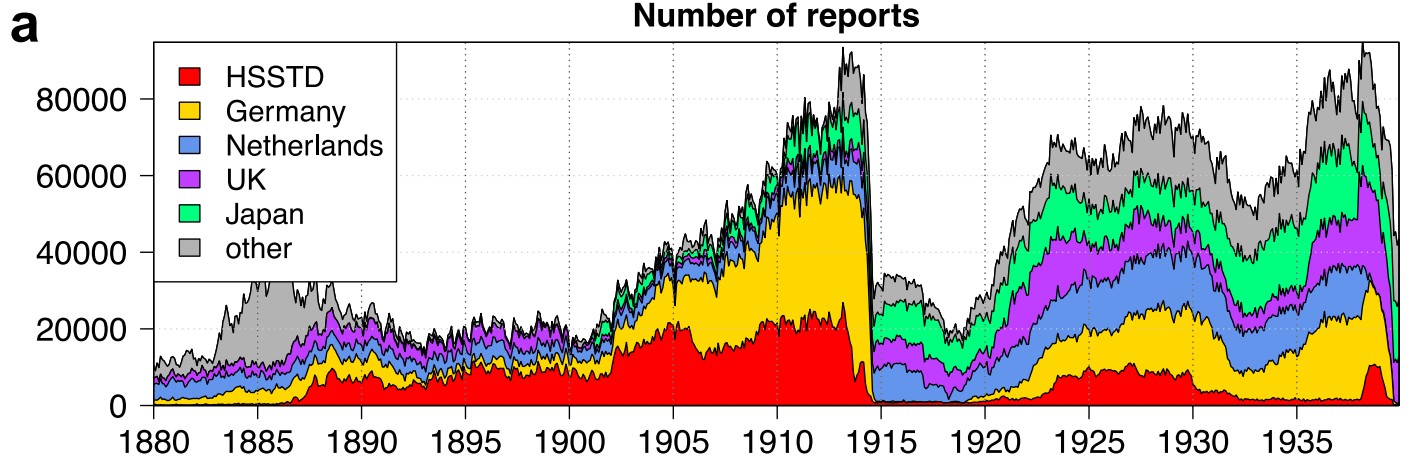

**a** Number of reports

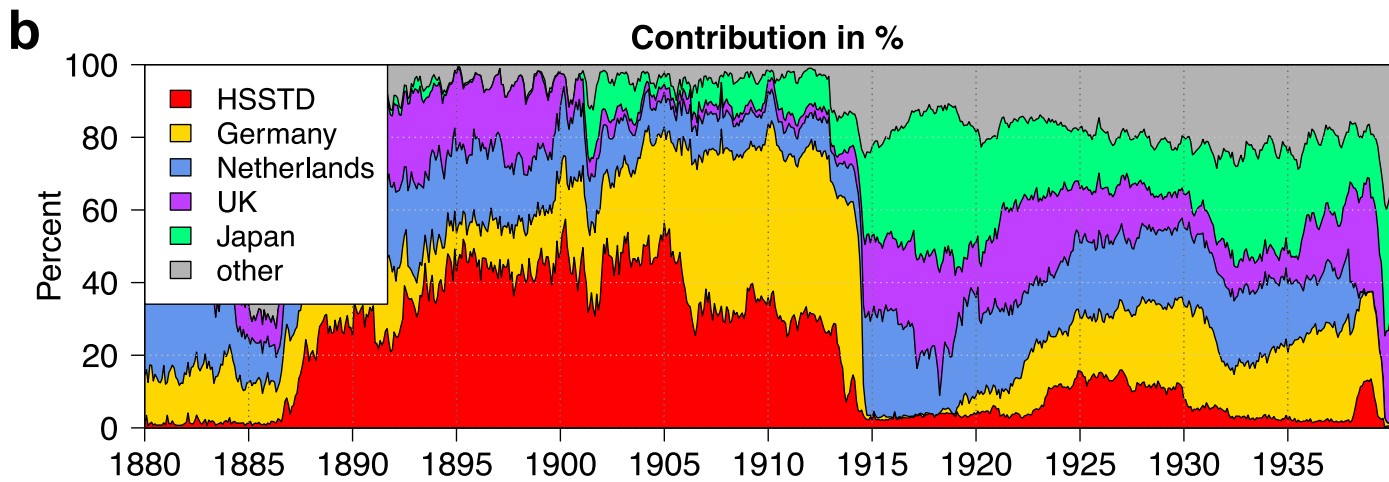

**b** Contribution in %

**Extended Data Fig. 7 | Breakdown of contributions by month to ICOADS subsets split by country of origin, where HSSTD refers to the Historical Sea Surface Temperature Data project[36]. a.** number of reports per month; **b.** as percentage of total.

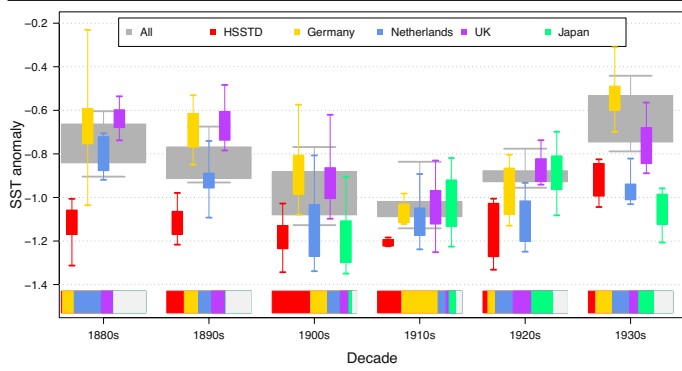

**Extended Data Fig. 8 | Summary of anomalies by decade for ICOADS subsets split by country of origin.** Boxes show interquartile range of global monthly mean anomalies in decades from 1880s to 1930s, whiskers indicate the full range. The wide grey boxes are for all observations and the colored boxes for contributing subsets as shown in the legend. The colored bars at the bottom show the fractional contribution of each subset to each decade with light grey representing "other" observations not included in the country subsets. Subsets are HSSTD (ICOADS DCKs 150-156, predominantly data from US); Germany (DCKs 192,215,720); Netherlands (DCK 193); UK (DCKs 194,201-204,216,245); Japan (DCKs 118, 762); see ref. 78. Prior to 1900 most of the reports classified as "other" come from DCK 704, later reports are largely from DCKs 705-707. Further analysis on ICOADS contributing sources is available in the Supplementary Information and relies on references38,79–82.