## [Peer Review File · Nature]

Manuscript Title: Early-twentieth-century cold bias in ocean surface temperature observations

Reviewer Comments & Author Rebuttals

Reviewer Reports on the Initial Version:

Referee #1 (Remarks to the Author):

Sippel et al.'s manuscript addresses the a major feature of the instrumental global temperature record, namely the relative coolness during 1900-1930 that is apparent in existing sea surface temperature (SST) datasets. Their key results demonstrate that the real-world SST during this multi-decadal is unlikely to have been as cold as current SST datasets suggest, and that these datasets are therefore have systematic errors during this period that have not previously been corrected or captured in existing statistical representations of SST errors (SST statistical error models). They establish these results through a systematic and sophisticated analysis of land and marine datasets and their association with global temperature. They then back up their findings with impressive supporting analyses such as comparison of coastal SST versus nearby coastal land air temperatures, comparison with non-instrumental (palaeoclimate) proxy reconstructions/records for the same periods, repeating the analysis with alternative instrumental datasets including marine air temperature, and an exploration of potential sources of error in the underlying database (ICOADS) of SST measurements.

The paper is original: no previous study has attempted such a comprehensive consideration of this problem, combining instrumental source data, instrumental data products, palaeoclimate data and climate model simulations. It really is a "tour de force". While, of course, this is not the final word on this problem, it does represent a very significant step forward and a robust step on which future work can be built.

The significance is set out by the authors at the end of the abstract and elsewhere in the paper. The major remaining discrepancy between climate model simulations of the historical instrumental period is this apparent cold anomaly in the instrumental record, and Sippel et al. have demonstrated that a considerable part of this model-obs discrepancy very likely arises from artefacts in the instrumental SST record rather than from deficiencies in climate models. This is important for provided a more complete explanation of historical warming and also for our understanding of natural, unforced, multidecadal variations in global climate. The role of purported multidecadal variability (e.g. from Atlantic and Pacific multidecadal variability, i.e. AMO/AMV and IPO/PDV) on global temperatures is likely to have been previously overstated according to this work, but without diminishing their importance in regional-scale climate.

The manuscript is comprehensive and sophisticated in its approach (in terms of the use of ridge regression trained and tested on thousands of years of climate model simulation, incorporating a representation of time-varying instrumental uncertainty and coverage, and also in terms of the variety of datasets/comparisons to which it is applied). Despite this, the authors have made a great job at expressing the approach and their findings (and the complementary analyses that they

performed to test the robustness of their findings) in a way that should (mostly) be understandable by researchers who are not working in this niche area. I say "mostly" because there are a few clarifications that could be made, covered in my detailed comments below.

I see no major problems with the work or its presentation. Its wide significance and the approachable way in which the work has been described make it very suitable for a journal like Nature. I therefore recommend that it be published in Nature.

Minor comments

1. *Title and some other locations.* Sometimes it isn't always clear that the finding of the paper is not that there was a cool period but that there wasn't. Despite the title having the word "apparent" it could easily be misread to imply that the paper is studying a genuine ocean cold anomaly. Similarly L193 says the cooling is likely too strong. Too strong for what? Too strong in what? Perhaps better to say "late 19th century cooling depicted by current global temperature datasets is likely stronger than in reality"? L217 says "agreement between models and observations may be higher than previously thought" but surely you mean "agreement between models and reality may be higher than current observational datasets suggest"? Go through it carefully and see if there are other examples of potential confusion on this issue.

2. *Conclusion section (or somewhere).* Training the statistical model to predict global-mean temperature (or other target metrics) is based on a large ensemble of CMIP6 models. As an overall ensemble, these do not simulate sustained (multidecadal) differences between land air temperature and SST anomalies and therefore the trained statistical models will not do so either, and this leads to one line of evidence for establishing that the instrumental SST datasets are incorrect in depicting a sustained relative cold period during 1900-1930. However, did you consider stratifying the CMIP6 ensemble into three groups according to the magnitude of the land-sea warming contrast that they simulate and training the ridge regression model on each of the three groups in turn? It would be interesting to know if the third of CMIP6 models that simulate the strongest land-sea warming contrasts might also support stronger multidecadal variability in land-sea temperature differences. This may not be feasible to do (e.g. if the resulting dataset is too small to train a robust model) or maybe it is already captured in the shaded ranges shown in Fig. 2.

3. *Conclusion section (or somewhere).* Is there a potential concern of circularity if future modifications to the global temperature record are implemented because of your work and result in better agreement with climate models, given that your work is partly based on climate models? Is this a concern? You might add a sentence to address this and explain why it isn't a concern.

4. L80: clarify what the correlation $R=0.47$ is based on. You just say "raw data"; is it the correlation between the published CRUTEM5 and HadSST4 series?

5. L96: this is the first time that "unadjusted HadSST4 data" is mentioned, but you don't explain what this means. A reader might think that it is unadjusted by you, i.e. it is the published HadSST4 data product. But I think you mean that it is HadSST4 if the adjustments already made in creating the HadSST4 data product had not been used?

6. L156-181: this is a useful exploration of SST data sources that may explain the uncorrected cold bias in existing SST datasets. But it makes no mention of the truncation to whole-degree Celsius when Japanese data were digitised (as explained in your ref 30): is this because you don't find it to be a significant contributor to the cold bias, or because it has already been taken into consideration in the SST datasets that you are using?

7. L195: "the constraints imply": up to now you haven't mentioned what the constraints are.

8. L204: "a prolonged cold period": do your results show if there is any seasonality in the implausible cold anomaly that might help to pin down its source?

9. Method P18 "Monthly coverage maps are taken from..." state if this sentence is talking about the coverage of *non-infilled* data.

10. End of P21, start of P22: this talks about the ensemble produced for HadCRUT5 non-infilled dataset and that this is used to represent the correlated uncertainties of the CRUTEM5 land air temperatures. However I don't think the land-only fields are provided by HadCRUT5 so the coastal grid cells will be influenced by SST too. So did you generate your own ensemble just for the land air temperature data, but following the method described by Morice et al. for HadCRUT5? You also say that there is no spatially correlated error term for the land temperatures, but are the urbanisation and exposure biases not spatially correlated in CRUTEM5/HadCRU5?

11. P23 1st para (and elsewhere where this is discussed): the discussion highlights that inclusion of observational uncertainty in training the ridge regression model is important because it affects the regression weights, and this is true (and a very useful additional finding of this paper). However the explanation here and in the supplementary info (SI) says that this is because it put less weight on grid cells that have larger uncertainties/biases. Are the cells that are downweighted SI Fig1 really more uncertain? Of course if there are some grid cells that are more uncertain then including that uncertainty in the training will indeed downweight them, but it looks like the main explanation is simply that including uncertainty (even if it were uniform uncertainty) would even out of the regression weights so that they are less focussed on a small number of grid cells and instead their predictive skill is spread across a greater number of grid cells.

12. P24: it mentions an "unseen model". Clarify that all initial condition ensemble members from this climate model have been withheld, so the model is definitely "unseen".

13. P24/P25: I couldn't understand the sentence that spans this page boundary. Are you again referring to models having differing numbers of initial condition ensemble members? Maybe refer to a section in the methods or SI where this is explained?

14. P27: "Second, we train on... with the global mean removed". But do you then apply the trained ridge regression model to SST data without its global mean removed? If not, then how do you get the trend in SI Fig7(a)? What does that trend represent?

15. P28: "In a transfer learning manner..." I couldn't understand this sentence. Maybe delete it and leave the SI to explain this point?

16. P32, first bullet point. Here (and elsewhere in the paper) it is not always clear if you have made a new datasets based on coastal comparisons or you are always using the one created by Cowtan et al. (2018; your ref 16)? Don't assume the reader knows what 'coastal hybrid SST' means or that the various ways you describe it (e.g. here, L100 'Cowtan-HybridSST', etc.) are describing the same dataset.

17. P32, second bullet point. "based on the linear relationship... shown in Fig 5". Do you mean from CMIP6 in Fig 5? Fig 5 shows other datasets too, so best to be 100% clear on this.

18. Fig1: hard to follow what panel (g) is showing, the caption is brief on this. Is this using the constraint from the linear CMIP6 relationship in Fig 5, together with the land data to predict the SST adjustment needed to bring them into agreement? Or something like that?

19. Fig2: caption needs to say what the grey and pink shading represent (95% ranges or full ranges?).

20. Fig3: caption should say what the symbols, horizontal black lines and shaded distributions are in panel (c).

21. Fig5: caption is too brief. What are the ellipses in (a)? Caption for (a) only mentions CMIP6 but it also shows observational data too. What do you mean by constraint in panel (b)?

22. SI Table 4: give a total for each column (number of models, number of training runs, number of analysis runs) so the reader can cross-match this to the numbers mentioned in the text.

Referee #2 (Remarks to the Author):

Dear Authors and Editors,

The submitted study provides numerous lines of evidence that there is a physically implausible discrepancy between global sea surface temperature (SST) and land surface air temperature (LSAT) during the early 20th century (1900-1930). The authors claim that this discrepancy in LSAT and SST arises due to a cold bias in SST observations because of poorly corrected biases in the direct observations of SST.

The consequences of the presented findings are far reaching and touch many key areas of basic and applied climate science. The authors mention 3 key consequences as (1) a decrease in the 20th century warming trend, a key IPCC statistic and measure of global change, (2) a decrease in the estimated global decadal variability in SST, a key metric for quantifying the internal variability of the climate system, and (3) a pathway towards improvement agreement between observed global temperature and CMIP simulations if the SST biases are corrected. If anything, the authors are underselling the broad significance of their findings. Global Mean Surface Temperature is arguably the most important metric of global climate as a measure of global energy imbalance and therefore the forced response as well as global internal variability. Nearly every subfield of climate science, climate impacts, and climate policy uses GMST frequently in analyses and GMST is prominently used in all IPCC reports and the Paris Agreement. The authors' suggestion that the GMST needs correcting is a major, novel finding with broad impact.

The study utilizes nearly all available tools to investigate the physicality of the "cold anomaly in the early twentieth century" and I am very impressed with the methods, statistics and uncertainty quantification, variety of data sources, and robustness of the findings. From my perspective, the key finding of the study is that there is a physically implausible bias in observed SST from 1900-1930 and that this bias is not present in the LSAT record. This finding is supported with observational, climate model, and paleoclimate analyses, all of which use the best practices for including uncertainty at each step in the analysis.

The majority of my following comments are about the presentation of the findings which should be improved before publication. The main text needs substantial work to make the findings and justifications clear to a broad readership. I would suggest defining key terms and periods explicitly and making clear how lines of evidence support their main points. I was always able to determine what the authors meant in a given sentence, but often only as a result of my knowledge of historical temperature data. I will provide my edits in the line-by-line comments section, but would suggest that the authors get a detailed friendly review from a climate scientist from a different subfield to make sure the paper is broadly accessible. I suggest this as the results are broadly exciting and want to ensure the full impact of the findings. In addition, Figure 1 is critical in explaining the problem to the reader, but is very hard to interpret and should be reworked to make clear the problem addressed by the study.

In sum, the submitted study is a rigorous analysis with clear conclusions that will have broad impact. I support the publication of this article after addressing the following comments.

Minor Comments:

(1) I suggest the authors revise the title of the manuscript to reflect that the cold anomaly is in the SSTs as well as that this cold anomaly is found in observations but believed to be erroneous due to observational biases.

(2) The various HadSST4 products are critical to the conclusions of the study and not sufficiently explained in the main text. As I understand, in order of proposed accuracy from worst to best, there are (1) HadSST4-unadj (2) HadSST4 and (3) The SST fields from Cowtan-HybridSST which has coastal corrections to SST from relevant LSAT measurements. I infer various points that are not made clear and are necessary for interpreting the presented results:

HadSST4-unadj does not include the bias corrections to observations that are included in operational HadSST4

HadSST4 is in fact operational HadSST4 following Kennedy et al. (2019)

Cowtan-HybridSST is a processed version of operational HadSST3 with corrections to SST following Cowtan et al. (2018).

We expect the accuracy of products to be ranked (1) to (3) from worst to best as we expect the correction of known biases in SSTs to improve the estimation of global mean temperature.

I would suggest that the authors outline these products early in the main text. I would also suggest naming Cowtan-HybridSST differently to make clear that it is most closely comparable to HadSST4.

(3) Related to comment (2), I have recently reviewed work by some of the authors on the manuscript (Chan et al. 202X) on a novel, bias-corrected SST product. I would suggest including this data in the study in some capacity if the editor approves. I recommend some inclusion of this product, even if limited by the less developed uncertainty assessment, as the work by Chan and colleagues represents the state of the art in SST bias corrections and I expect any analyses using this product will further support the presented conclusions.

(4) Figure 1 is currently very difficult to read and should be redesigned. The colors do not allow distinction between HadSST4-unadj and Cowtan-HybridSST or CRUTEMP5 and BEST-Land without context. In my view, the most powerful comparisons here are CRUTEMP5 with the three SST products discussed in comment (2). Thus, I would suggest that the authors simplify panels a-f in some manner by either removing or deemphasizing other series. The other series do provide useful context and support for the findings, but currently serve more as a distraction.

Another potential way to make the figure more impactful is to provide a zoom in to the time period of interest. One panel could show the entire record, but the remainder could focus on 1870-1950, for example. This could help highlight the time period of interest and make the figures more interpretable. If you do not do a zoom in, it could be helpful to make key time periods that are referred to in the text with faint horizontal lines to align the reader to the interesting time periods.

Finally, make clear that the implied SST adjustments are with respect to HadSST4-unadj in the subfigure title.

(5) There are various time periods discussed in this study, but they are not always clearly defined in the text. I would suggest always writing about a time period by using the years of the period and sticking to as few of these periods as possible to improve continuity. These periods could be denoted visually in key figures as well to better connect the text and the figures as suggested in comment (4) for Figure 1.

(6) The section on “Marine data sources during the ocean cold anomaly” at L156 felt out of place and longer than necessary as it doesn’t present any findings. I would suggest summarizing this section in a sentence or two in the introduction and/or conclusion. The reclaimed words would be useful in more completely spelling out the presented findings.

(7) The language used to describe the ridge regression model is awkward at times and not consistent throughout the text and methods. I would suggest using either the word “predict” or “estimate” to signal when ridge regression is being used to predict a time series from a field. This will clean up the main text as well as dramatically improve the readability of the methods. Also, make clear what language you are going to use for your target time series or predictand. At times, the word “source” is used which is non-standard and confusing.

Line-by-line Comments

L25: Make clear here that you show that the SST cold anomaly is unforced and that IV can’t explain. This is a super exciting finding of the study that could be better highlighted here.

L40: It would be helpful to orient the reader that there are three difficulties in assessing the quality and consistency of SSTs/LSATs before going into detail: (1) Observational biases (2) Interpolation/Extrapolation (3) The complexity of the climate system.

L49: This sentence leads me to ask why the LSAT record is judged to be more reliable. Either provide a short explanation or make this claim stronger if supported by previous studies.

L51-53: The sentence beginning with “research focused on filling these data gaps” felt out of place. I’m not sure the goal of this sentence here. Furthermore, most operational products do the infilling separately for LSAT and SST so I would somewhat disagree with the second half of the sentence even if new methods have focused on the full global field.

L66-67: Make clear that this is the “statistical model” as you were just discussing CMIP6 climate models.

L98 “extended archive” of what?

L109: I would suggest a paragraph break before “Next, we focus on attribution”. Also, I would suggest clearly spelling out that you have shown in Figures 1d-f that the forced response and interannual variability cannot explain the difference between ocean-derived estimates, leaving the decadal IV as the discrepancy. You do say this, but taking a few more lines to really make this clear will make the next analysis more impactful to the reader.

L116: replicate -> replicates

L118: I suggest adding “[SST/LSAT] reconstructions during the ocean...”

L121: Should be Fig2d and Fig 2e?

L124: It could be helpful here to include a sentence explicitly stating that your CMIP analysis suggests that the cold anomaly cannot be explained by either the CMIP6 forced response or internal variability to really drive this point home. You have shown this here, but don’t directly say it.

L143: I suggest removing “upper range” unless supported by values. To eye, they appear to simply “align”

L144-145: It is not clear what is meant by the second half of the sentence, “highlighting a discrepancy...”

L183: Make clear that you are talking about the 1900-1930 or “early 20th century” cold anomaly again here.

L184: Briefly Introduce the ETCW here or earlier in the text as it is not a ubiquitously known problem.

L187-190: It could be clarifying/helpful to edit or append this sentence to put this finding in the context of external forcing vs. internal variability.

L202: Should this be “global temperature [interannual] variability” as there is not agreement on decadal scales as discussed?

L216: It is not clear what “constraints” refers to.

Last sentence: The last sentence should be much stronger. The presented study has provided multiple lines of robust evidence that the current SST record is incorrect in the early 20th century and the closing sentence, and conclusion, should reflect this.

Methods p21: It appears the three components are treated as statistically independent. If this is the case, make this clear when introducing them.

Methods p22: How are the uncorrelated uncertainties estimated with gridded error fields?

Methods p27: I suggest adding a paragraph break before “In addition”

Methods p27: The description of the additional reconstructions is not clear. I would suggest editing this entire section following my suggestion in minor comment (7).

Methods p31: Define “Turn of the 20th century” in years in the title.

Figure 4: It could be helpful to add a horizontal line at 0 to provide a reference for readers to compare distributions in both panels a and b.

Author Rebuttals to Initial Comments:

We thank the reviewers for the careful, knowledgeable and positive evaluation of our manuscript, and we acknowledge the Editorial suggestions. We have carefully revised the manuscript, following the suggestions we received. We provide detailed responses to all comments in this document. In the following the reviewer comments appear in black with the author responses in blue.

Referees' comments:

Referee #1 (Remarks to the Author):

Sippel et al.'s manuscript addresses the a major feature of the instrumental global temperature record, namely the relative coolness during 1900-1930 that is apparent in existing sea surface temperature (SST) datasets. Their key results demonstrate that the real-world SST during this multi-decadal is unlikely to have been as cold as current SST datasets suggest, and that these datasets are therefore have systematic errors during this period that have not previously been corrected or captured in existing statistical representations of SST errors (SST statistical error models). They establish these results through a systematic and sophisticated analysis of land and marine datasets and their association with global temperature. They then back up their findings with impressive supporting analyses such as comparison of coastal SST versus nearby coastal land air temperatures, comparison with non-instrumental (palaeoclimate) proxy reconstructions/records for the same periods, repeating the analysis with alternative instrumental datasets including marine air temperature, and an exploration of potential sources of error in the underlying database (ICOADS) of SST measurements.

The paper is original: no previous study has attempted such a comprehensive consideration of this problem, combining instrumental source data, instrumental data products, palaeoclimate data and climate model simulations. It really is a "tour de force". While, of course, this is not the final word on this problem, it does represent a very significant step forward and a robust step on which future work can be built.

The significance is set out by the authors at the end of the abstract and elsewhere in the paper. The major remaining discrepancy between climate model simulations of the historical instrumental period is this apparent cold anomaly in the instrumental record, and Sippel et al. have demonstrated that a considerable part of this model-obs discrepancy very likely arises from artefacts in the instrumental SST record rather than from deficiencies in climate models. This is important for provided a more complete explanation of historical warming and also for our understanding of natural, unforced, multidecadal variations in global climate. The role of purported multidecadal variability (e.g. from Atlantic and Pacific multidecadal variability, i.e. AMO/AMV and IPO/PDV) on global temperatures is likely to have been previously overstated according to this work, but without diminishing their importance in regional-scale climate.

The manuscript is comprehensive and sophisticated in its approach (in terms of the use of ridge regression trained and tested on thousands of years of climate model simulation, incorporating a representation of time-varying instrumental uncertainty and coverage, and also in terms of the variety of datasets/comparisons to which it is applied). Despite this, the authors have made a great job at expressing the approach and their findings (and the complementary analyses that they performed to test the robustness of their findings) in a way that should (mostly) be understandable by researchers who are not working in this niche area. I say "mostly" because there are a few clarifications that could

be made, covered in my detailed comments below.

I see no major problems with the work or its presentation. Its wide significance and the approachable way in which the work has been described make it very suitable for a journal like Nature. I therefore recommend that it be published in Nature.

We thank the reviewer for the positive evaluation of our work, and for the very insightful comments. We address the in-depth comments below.

Minor comments

1. *Title and some other locations.* Sometimes it isn't always clear that the finding of the paper is not that there was a cool period but that there wasn't. Despite the title having the word "apparent" it could easily be misread to imply that the paper is studying a genuine ocean cold anomaly. Similarly L193 says the cooling is likely too strong. Too strong for what? Too strong in what? Perhaps better to say "late 19th century cooling depicted by current global temperature datasets is likely stronger than in reality"? L217 says "agreement between models and observations may be higher than previously thought" but surely you mean "agreement between models and reality may be higher than current observational datasets suggest"? Go through it carefully and see if there are other examples of potential confusion on this issue.

Thank you for these comments, which are very consistent with the feedback we received from reviewer #2 and the Editor. We have revised the title of our paper to reflect this point, that is to say more clearly that this cold anomaly is not consistent with the other datasets and lines of evidence. Our revised title reads: "Early twentieth century cold bias in ocean surface temperature observations". We have included your suggestions (in what was previously L193 and L217; now L239 and L272) to make this point even more explicit.

Also in the summary, we have tried to convey this point more unambiguously. We state that the cold *anomaly* we find in the reconstruction, is shown to be inconsistent with known forcings or natural variability, resulting in the likely conclusion that it is a *bias* in observations, L 25-29.

2. *Conclusion section (or somewhere).*

Training the statistical model to predict global-mean temperature (or other target metrics) is based on a large ensemble of CMIP6 models. As an overall ensemble, these do not simulate sustained (multidecadal) differences between land air temperature and SST anomalies and therefore the trained statistical models will not do so either, and this leads to one line of evidence for establishing that the instrumental SST datasets are incorrect in depicting a sustained relative cold period during 1900-1930. However, did you consider stratifying the CMIP6 ensemble into three groups according to the magnitude of the land-sea warming contrast that they simulate and training the ridge regression model on each of the three groups in turn? It would be interesting to know if the third of CMIP6 models that simulate the strongest land-sea warming contrasts might also support stronger multidecadal variability in land-sea temperature differences. This may not be feasible to do (e.g. if the resulting dataset is too small to train a robust model) or maybe it is already captured in the shaded ranges shown in Fig. 2.

It is a really intriguing question to ask whether the land-ocean warming ratio across models is related to the *magnitude of land-ocean differences* on multidecadal time scales. If this were the case, it would open new opportunities for emergent constraints; and it would be indeed potentially very useful to consider such model differences in the training setup.

A recently finished Master thesis in the Climate Physics group at ETH Zurich (Master Student Julia Dworzak, Title: “Understanding and constraining the land-ocean warming contrast’s inter-model spread”, supervised by E. Fischer and S. Sippel) had addressed this question and found that it is very difficult to find robust relationships on the inter-model spread in the land-ocean warming ratio, partly because of different responses to forcing across models (e.g., aerosol forcing in transient simulations) as well as internal variability. To address the reviewer’s comment, we briefly reassess this question by testing whether the suggested relationship exists (given the coverage masks of the late 19th and early twentieth century). Please see our Response Figure 1: There is a very strong relationship, across models and individual simulations between land warming and ocean warming trends for the 1980-2014 period (top panel), and also for other historical periods (see Fig. 5 in the main text). Yet, if we relate the land-ocean warming ratio to the magnitude of the 1901-1930 ocean-land difference (computed similar to Fig. 2b by considering the predictions of our statistical model), there is almost no relationship across models and simulations (middle panel). If we aggregate by model by considering the standard deviation of these differences (for each model we consider all its ensemble members and all 30-year periods before 1950 to increase sample size), we again find almost no relationship between the forced land-ocean warming ratio and the variability in the land-ocean difference (bottom panel). Hence we conclude that we can at the moment unfortunately not identify such a relationship.

Yet, the reviewer is of course correct in that structural differences across CMIP6 models (such as differences in variability patterns, etc.) could affect the estimated coefficients of our regularized linear model, which in turn could lead to differences in the extent to which individual model’s variability patterns (in land or ocean) would project onto those coefficients. In an earlier paper, written by a subset of the authors, we have tested different setups and CMIP training splits extensively (Sippel et al., 2021, *Science Advances*, doi:[10.1126/sciadv.abh4429](https://doi.org/10.1126/sciadv.abh4429)), where the goal was to understand how to best estimate the forced response while minimizing the extent to which patterns of decadal variability project onto the statistical model’s forced response. In the present case, we have experimented with different training setups initially, but found that for the problem of estimating global mean temperature from an incomplete coverage (rather than estimating the forced response), the training setup made virtually no difference, as the regression coefficients are smooth in space (and are linear) in all cases. Furthermore, as we train two independent models for the land- and ocean-based reconstruction, the statistical model is not optimized in any way for land-ocean differences. The smooth spatial regression coefficients and the fact that the statistical model is not optimized on land-ocean differences implies that the CMIP models’ land and ocean variability will be translated linearly into the GMST estimate, and hence decadal variability in the land/ocean patterns will be captured (that is, the linear model should maintain a large or small land-ocean difference from any CMIP model that produces such a pattern). We do in fact observe this as the observational datasets show a large-multi-decadal difference, and the CMIP6 models show variation in this metric (Response Figure 1, bottom panel), but not such large differences as part of their simulated multidecadal variability.

Response Figure 1: (top) Correlation between land warming and ocean warming is well constrained across models. (middle) Little correlation between land-ocean warming ratio and the land-ocean difference. (bottom) Standard deviation of the magnitude of the ocean-land differences, grouped by model.

3. *Conclusion section (or somewhere).* Is there a potential concern of circularity if future modifications to the global temperature record are implemented because of your work and result in better agreement with climate models, given that your work is partly based on climate models? Is this a concern? You might add a sentence to address this and explain why it isn't a concern.

Thank you, this is indeed a very important point that has to be carefully considered and explained. Indeed, if the SST data would be “naively” corrected to match the land data, there would be a risk of circularity one needs to be aware of, which is also discussed in Cowtan et al. (2018; doi:10.1002/qj.3235). Hence, additional constraints are necessary.

The main contribution of our paper is in highlighting the early 20th century discrepancy between the land and ocean record, and with several lines of evidence we show that a large part of the problem is very likely in the SST record. There are multiple historical examples, where models have been used to assess consistency of datasets, and can inform improvements:

- the observed early 20th century warming (ca. 1900-1945) was explained by some models by solar forcing (Meehl et al., 2004), even though studies later pointed to large uncertainties in solar forcing reconstructions and attribution studies indicated a small effect (Hegerl et al., 2018; doi:10.1002/wcc.522). A paper by Thompson et al. (2008; doi:10.1038/nature06982) pointing to the inconsistency of the SST data just after WW2, and later papers confirming SST inconsistencies during WW2 and correcting them (Chan et al. 2018; doi:10.1175/JCLI-D-20-0907.1);
- An apparent ocean cooling observed by Lyman et al. (2006), which was inconsistent with energy budget considerations and models, was later found to be a problem with ARGO float measurements;
- the large discrepancy between observed upper atmosphere temperature trends and simulated trends around 2005; assessed in the CCSP report by Karl et al. (2005), led to the identification of problems with radiosondes and satellite MSU data (Fu et al., 2004; Sherwood et al., 2005; Allen and Sherwood, 2008).

We hope that our paper will motivate new investigations into raw SST data and data rescue projects, thus motivating a variety of different approaches to correcting SSTs. Beyond that, our analysis of in-situ SSTs shows that different sources of SSTs exhibit different offsets. These offsets are understandable in terms of known measurement characteristics and show spatial and temporal variations that cannot be properly adjusted using the bias adjustment models employed in the products underpinning the most recent IPCC assessment. This information on different biases among sources is completely independent of the climate models.

To address your concern, we have included a sentence in the conclusion, which emphasizes the need for a variety of SST bias correction approaches (L265-270): “Continued rescue activities of historical climate data and metadata (Brunet et al. 2011), as well as the development of different SST bias correction approaches (Kent et al., 2017) are therefore crucial to better understand this key period, including its regional and seasonal characteristics. Such diverse bias correction approaches, including group-based SST adjustments (Chan et al., 2019), constraints from the SST diurnal cycle (Carella et al. 2018), from coastal (Chan et al., 2023) and paleoclimate data (Pfeiffer et al., 2017), will help ensure that any modifications to the global temperature record are robust.”

4. L80: clarify what the correlation $R=0.47$ is based on. You just say "raw data"; is it the correlation between the published CRUTEM5 and HadSST4 series?

Yes, it is the correlation between the published global mean time series of CRUTEM5 and HadSST4 in the respective period, and we clarified this in the revised manuscript (L. 97-98).

5. L96: this is the first time that "unadjusted HadSST4 data" is mentioned, but you don't explain what this means. A reader might think that it is unadjusted by you, i.e. it is the published HadSST4 data product. But I think you mean that it is HadSST4 if the adjustments already made in creating the HadSST4 data product had not been used?

Yes, that is correct and we agree it is important to avoid this potential misunderstanding. We have clarified this point by adding in (now) L. 86: "[...] in unadjusted HadSST4 data, that is the raw gridded data prior to any applied corrections."

In addition, we have streamlined the terminology of all datasets across the whole paper's text, figures, and tables, as requested also by Reviewer #2 (see our reply to R2).

6. L156-181: this is a useful exploration of SST data sources that may explain the uncorrected cold bias in existing SST datasets. But it makes no mention of the truncation to whole-degree Celsius when Japanese data were digitised (as explained in your ref 30): is this because you don't find it to be a significant contributor to the cold bias, or because it has already been taken into consideration in the SST datasets that you are using?

Thanks for highlighting this, it is another important point we need to clarify: Chan et al., 2019 derived a corrected SST dataset based on a mixed effects model framework (based on HadSST3), which identifies systematic offsets between different decks of SST observations. They identified important biases both in the Atlantic and in the Pacific, which however largely offset each other globally. The mixed effect model methodology, however, is not able to identify global-scale biases, as it only identifies relative offsets between decks. Consequently, the global mean change between the standard HadSST3 and the revised version by Chan et al. (2019) is negligible (see Extended Figure 8 in their paper). In an earlier stage of our study, we had tested whether we get a different reconstruction result when using the Chan et al. (2019) corrected SST dataset, but the results of our reconstruction were virtually identical so we decided to not include it in the paper to keep the results and presentation as simple as possible.

Specifically on the Japanese data truncation: The Japanese data affected by the truncation bias (the correction of which led to an increased and more uniform warming in the early 20th century in Chan et al. (2019)) only starts around 1925, as seen in our Supplementary Figure 9: Post-1930 the Japanese data are relatively cold due to the truncation, and the German data are relatively warm, but this is when coming out of the overall cold bias period. We have revised the section that discusses SST biases (L186-218) to now also account for the comments made by reviewer #2. We mention that "large and rapidly changing differences between observations from national fleets contributing data in different regions of the ocean cannot be accounted for using the common bias-correction methods based on globally-constrained and slowly-varying adjustment fields, and are hence likely to induce residual biases." (L199-203); which includes digitalization biases implicitly with the other possible biases linked to the distinct SST decks (or countries).

7. L195: "the constraints imply": up to now you haven't mentioned what the constraints are.

We have revised the manuscript such that it now properly introduces the constraints in the Main Text (L233-238): "multi-decadal land warming is in fact tightly coupled to ocean warming according to physical theory (Sutton et al., 2007; Byrne et al. 2018), and for the remainder of the observational

record. Here, we exploit this relationship and derive observational constraints on ocean warming from our land warming reconstruction; and further constraints from coastal hybrid SST data and the PAGES 2k and Ocean2k paleoclimate reconstructions (Neukom et al., 2019; Tierney et al., 2015) using the method of emergent constraints (Cox et al., 2013; details see Methods). Given these constraints, [...]"

We have also added a comprehensive explanation on how the constraints are derived in the method section (p. 40-41), which we agree was too short in the submitted version.

8. L204: "a prolonged cold period": do your results show if there is any seasonality in the implausible cold anomaly that might help to pin down its source?

We had checked that initially and we found only a relatively minor seasonal effect in the cold anomaly. While the seasonal effect is clearly interesting and may help to pin down the origin of the cold anomaly, we think that with our *global* reconstructions it may not be the most straightforward way to look at seasonality. Folland and Parker (1995) did a lot of work looking at the seasonal cycles as evidence of a change in character of the observations, but did so regionally. They showed that seasonal cycles were larger in the period when buckets were in use, and used that to tune their adjustment fields, which were then weighted so the trend was consistent with the trend in night-time marine air temperatures.

Hence, there is a lot of evidence that bucket measurements (and different types of buckets as well) show enhanced diurnal and seasonal cycles, but doing an analysis robust enough to add something to this paper would be out of scope as the global analysis is not directly suitable to look at seasonality. However, this may be a clear way forward, and so we have added a short sentence in this direction (L265-270): "Continued rescue activities of historical climate data and metadata (Brunet et al. 2011), as well as the development of different SST bias correction approaches (Kent et al., 2017) are therefore crucial to better understand this key period, including its regional and seasonal characteristics. Such diverse bias correction approaches, including group-based SST adjustments (Chan et al., 2019), constraints from the SST diurnal cycle (Carella et al. 2018), from coastal (Chan et al., 2023) and paleoclimate data (Pfeiffer et al., 2017), will help ensure that any modifications to the global temperature record are robust."

9. Method P18 "Monthly coverage maps are taken from..." state if this sentence is talking about the coverage of *non-infilled* data.

Yes, it does. Fixed.

10. End of P21, start of P22: this talks about the ensemble produced for HadCRUT5 non-infilled dataset and that this is used to represent the correlated uncertainties of the CRUTEM5 land air temperatures. However I don't think the land-only fields are provided by HadCRUT5 so the coastal grid cells will be influenced by SST too. So did you generate your own ensemble just for the land air temperature data, but following the method described by Morice et al. for HadCRUT5? You also say that there is no spatially correlated error term for the land temperatures, but are the urbanisation and exposure biases not spatially correlated in CRUTEM5/HadCRU5?

You are correct: HadCRUT5 does not provide land-only fields for coastal grid cells; but HadCRUT5 provides the blended spatio-temporal ensemble fields and HadSST4 provides ocean-only ensemble

fields that are used for blending with CRUTEM5 ensemble fields in HadCRUT5. Hence, we inferred the CRUTEM5 spatio-temporal ensemble fields by inverting the blending methodology described in Morice et al., (2021), Section 3.4.1: Blending of CRUTEM5 and HadSST4 to derive HadCRUT5 is based on the formula

$$T(s, t) = f(s, t)T_L(s, t) + (1 - f(s, t))T_M(s, t),$$

with land area fraction weights $f(s, t)$, resulting in the blended HadCRUT5 temperature $T(s, t)$ for each ensemble member. $T_L(s, t)$ and $T_M(s, t)$ represent land and ocean temperatures, respectively. Hence, because ensemble members are available for $T(s, t)$ (from HadCRUT5) and for $T_M(s, t)$ (from HadSST4), we can infer the missing CRUTEM5 land-only ensemble fields via the corresponding formula also for coastal grid cells

$$T_L(s, t) = \frac{T(s, t) - (1 - f(s, t))T_M(s, t)}{f(s, t)}.$$

We have verified that the ensemble members in HadSST4 correspond to the same ensemble members in HadCRUT5.

Regarding urbanization and exposure biases: These biases do have a spatial and temporal structure and are encoded in the ensemble realizations of the CRUTEM5 ensemble according to Morice et al (2021): “The systematic effects of residual urbanization errors (Brohan et al., 2006; Parker, 2010) and nonstandard sensor enclosures (Folland et al., 2001; Parker, 1994) are sampled and encoded into the gridded ensemble at a regional level, again following the method of Morice et al. (2012)”. This point is clarified in the manuscript on p. 29 (Methods). The ensemble encapsulates uncertainties such as station-based homogenization errors and uncertainty in climatological normals, as well as regional urbanization errors and nonstandard sensor enclosure.

11. P23 1st para (and elsewhere where this is discussed): the discussion highlights that inclusion of observational uncertainty in training the ridge regression model is important because it affects the regression weights, and this is true (and a very useful additional finding of this paper). However the explanation here and in the supplementary info (SI) says that this is because it put less weight on grid cells that have larger uncertainties/biases. Are the cells that are downweighted SI Fig1 really more uncertain? Of course if there are some grid cells that are more uncertain then including that uncertainty in the training will indeed downweight them, but it looks like the main explanation is simply that including uncertainty (even if it were uniform uncertainty) would even out of the regression weights so that they are less focussed on a small number of grid cells and instead their predictive skill is spread across a greater number of grid cells.

This is an interesting point. The reviewer is correct that only adding *uniform uncertainty* would result in the ridge penalization itself - that is, we get a scaled identity matrix as covariance of the noise which is just the ridge penalty, and we would hence choose a larger lambda and thus “wash out” the regression coefficients more uniformly in space in that case.

Yet, the biases (and uncertainties) are not uniform, and in particular the biases are correlated temporally. Hence, we focus on the biases and investigate whether the change in the regression

coefficients when adding the biases and uncertainties (for our illustration time step of June 1895 shown in Extended Data Figure 2-4) leads to a nonuniform change. To do that, we divide the grid cells into four different groups (four quartiles of biases ranging from a small standard deviation across the bias ensemble (Q1) to a large SD in the bias ensemble (Q4)) and we look at the L2 norm for each of the four groups, as the L2 norm tells us how large the regression coefficients are (in a sum of squared coefficients sense with the same number of grid cells in each group). In Response Figure 4 (left panel), we find that if no biases/uncertainties are added, the L2 norm is large in grid cells with large biases (Q3 and Q4 group); while the L2 norm is rather small in regions with small biases (Q1 and Q2). This is probably because grid cells with small biases are located primarily in North America, Europe and parts of Australia, which receive rather small regression coefficients. When biases/uncertainties are added during the training to the statistical model, we find that the L2 norm decreases overall as expected. Yet, if we compute the ratio between the L2 norm of the model with and without biases/uncertainties added (Response Fig. 4, right panel), we find that the regions with small biases are upweighted (in Bias Q1 a 2x increase in the L2 norm), while in Bias Q4 the grid cells are downweighted overall (relative change of 0.39). Hence, we argue that these nonuniform biases/uncertainties are indeed affecting the grid cells in a non-uniform way.

Response Figure 2: (left panel) L2 norms for the statistical model with and without biases/uncertainties included during training, stratified into four groups by the magnitude of the standard deviation of the biases in CRUTEM5. (right panel) Relative L2 norm change between the two statistical models (bias incl. vs. no bias) for each of the four bias quartile groups.

12. P24: it mentions an "unseen model". Clarify that all initial condition ensemble members from this climate model have been withheld, so the model is definitely "unseen".

Yes, the cross-validation is performed based on a "leave-one-model-out" approach (training on $k-1$ statistical models when k is the total number of models), that is we identify the optimal λ parameter based on unseen models only (including performance evaluation). Note that the final set of regression coefficients is derived as the average across all k statistical models.

13. P24/P25: I couldn't understand the sentence that spans this page boundary. Are you again referring to models having differing numbers of initial condition ensemble members? Maybe refer to a section in the methods or SI where this is explained?

Yes, we are referring to the fact that some models provide more ensemble members than others. In the training step of our statistical model, we want to avoid that the regression coefficients are overproportionally influenced by a climate model that just provides more ensemble members in CMIP6. Therefore we weight our regression such that each climate model receives equal weight for the extraction of the regression coefficients. We have clarified this point in the manuscript (p. 32, bottom): “Because several climate models provide different numbers of ensemble members, we weight the regression such that each climate model receives equal weight for the extraction of regression coefficients.”

14. P27: "Second, we train on... with the global mean removed". But do you then apply the trained ridge regression model to SST data without its global mean removed? If not, then how do you get the trend in SI Fig7(a)? What does that trend represent?

Indeed, we train a new model with the global mean removed from the predictors (from each time step, that is monthly temperature map, separately removed), but the target of the reconstruction remains the same. Hence, the statistical model “sees” only relative changes in the SST pattern over time (e.g., stronger Arctic warming, slow Southern Ocean warming relative to global SST warming, etc.). For example, in an earlier paper, Sippel et al. (2020), we had used a similar “global mean subtraction” approach to estimate the forced response from land+ocean data with no global mean change - in that case, the trend can be well captured because the statistical model learns the land-ocean warming contrast signatures (relative to the global mean temperature change, land is warming and ocean is cooling). Here, the “global mean subtraction” from SSTs clearly leads to an underestimation of the global trend, because we have removed an important part of the trend signal; but importantly, some of the recent trend is still captured: The statistical model “sees” the recent climate change trend in nuanced changes in the global SST pattern, and therefore still captures part of the recent trend. Interestingly, in the 1900-1940 period, only a very small global trend is captured, which indicates that the strong trends in the SST datasets in this period are not arising largely due to known patterns of variability captured in the models, as otherwise we should still see some trend emerge similar to the more recent period.

15. P28: "In a transfer learning manner..." I couldn't understand this sentence. Maybe delete it and leave the SI to explain this point?

We have deleted this statement, it was indeed confusing, and now we just say that the machine learning tool from Kadow et al. (2020) is used to infill the HadSST4 / CRUTEM5 datasets.

16. P32, first bullet point. Here (and elsewhere in the paper) it is not always clear if you have made a new datasets based on coastal comparisons or you are always using the one created by Cowtan et al. (2018; your ref 16)? Don't assume the reader knows what 'coastal hybrid SST' means or that the various ways you describe it (e.g. here, L100 'Cowtan-HybridSST', etc.) are describing the same dataset.

Thank you for pointing this out, we agree this was not clear in the submitted manuscript. We implemented several revisions:

- We now introduce the three main SST datasets (HadSST4, HadSST4-unadj, and CoastalHybridSST) and the nighttime marine air temperature data (ClassNMAT) early on in the paper (following a similar comment by R#2; L85-89); and we have streamlined the terminology between the figures and the text.
- In our submitted manuscript, we had shown the reconstruction made from the CoastalHybridSST dataset (i.e. from Cowtan et al., 2018) in Fig. 1 (without any correction). However, in Fig. 5 we had “corrected” the ocean cold anomaly in HadSST4 by merging with the low-frequency band from CoastalHybridSST in the extended period of the cold anomaly (brown color in previous Fig. 5a). However, we realize that there is a risk that this may lead to confusion, and therefore we have revised the figure to **only** use the HadSST4 reconstruction individually (dark grey crosses in new Fig. 5a), and **only** use the CoastalHybridSST reconstruction individually (brown crosses in new Fig. 5a). Hence, throughout the paper we now use HadSST4 and CoastalHybridSST reconstructions individually, but **not a combination** of the two. We have adjusted the text around p. 40-41 in the revised manuscript. This change has led to a minor change in the constrained global SST trend range in Fig. 5 for the CoastalHybridSST dataset: While we think that attempting a correction as done in the submitted version may be useful, we think that for the current paper it may be more easily understandable to just show the constrained range for HadSST4 and CoastalHybridSST individually.

17. P32, second bullet point. "based on the linear relationship... shown in Fig 5". Do you mean from CMIP6 in Fig 5? Fig 5 shows other datasets too, so best to be 100% clear on this.

We realize this has been insufficiently explained in the submitted manuscript. We added a comprehensive explanation on how the emergent constraint technique (Cox et al., 2013) is used (P40-41, “Derivation of observational constraints on historical ocean warming in Fig. 5”), and we have carefully revised Fig. 5:

For each large-scale temperature trend metric (predicted GMSST from HadSST4 and CoastalHybridSST, GMST for PAGES2k, predicted GMLSAT for CRUTEM5 and tropical mean SST for Ocean2k), we obtain the relationship between the respective predictor metric and our target metric (GMSST) across CMIP6 historical simulations, which all show strong linear relationships (as expected). We use the Cox et al. (2013) emergent constraint linear regression technique to derive *prediction intervals* for the constrained ranges for each of these datasets, thus accounting for internal variability in the uncertainty of the constrained ranges. In the submitted manuscript, we had shown the trends from the reconstructions for CRUTEM5 and HadSST4, but we realize it may be more useful to show the constrained ranges exactly as for the other datasets (hence *prediction intervals*) such that uncertainties are comparable. Hence, our revised Fig. 5 has changed compared to the submitted version, but we do think the changes are justified as we now **better account for uncertainty due to internal variability in the constrained ranges for all datasets** (the conclusions are identical). The prediction intervals are obtained following Cox et al. (2013), explained on p. 40-41.

18. Fig1: hard to follow what panel (g) is showing, the caption is brief on this. Is this using the constraint from the linear CMIP6 relationship in Fig 5, together with the land data to predict the SST adjustment needed to bring them into agreement? Or something like that?

We have clarified the title of panel (g) and the caption: Panel (g) shows the difference between our global reconstructions of HadSST4 and HadSST4-unadj, and the difference of CoastalHybridSST and HadSST4-unadj. Hence, panel (g) shows the (global mean) adjustments made to the raw, unadjusted HadSST4 data if HadSST4 or the CoastalHybridSST datasets are used.

19. Fig2: caption needs to say what the grey and pink shading represent (95% ranges or full ranges?).

We show 95% ranges (2.5th to 97.5th percentile) in Fig. 2. We have clarified this in the caption.

20. Fig3: caption should say what the symbols, horizontal black lines and shaded distributions are in panel (c).

Done. In addition, we have revised panel b to omit the stippling. It used to show the observed areas in HadSST4, but by careful reconsideration, we think it is rather confusing to show stippling based on HadSST4 coverage on top of the Berkeley Earth dataset. Information on HadSST4 coverage is available in panel (a).

21. Fig5: caption is too brief. What are the ellipses in (a)? Caption for (a) only mentions CMIP6 but it also shows observational data too. What do you mean by constraint in panel (b)?

We have clarified the ellipses in (a), and now we mention the observations, too. In (b), we now explain that we are using the method of emergent constraints to derive these estimates of global ocean warming. Note that we have added a comprehensive explanation on the emergent constraints in the methods, and also Fig. 5 has changed slightly because of the improved treatment of uncertainties: We now say in the caption explicitly that all error bars reflect 95% prediction intervals, thus accounting for uncertainty due to internal variability in the emergent constraints.

22. SI Table 4: give a total for each column (number of models, number of training runs, number of analysis runs) so the reader can cross-match this to the numbers mentioned in the text.
done.

Referee #2 (Remarks to the Author):

Dear Authors and Editors,

The submitted study provides numerous lines of evidence that there is a physically implausible discrepancy between global sea surface temperature (SST) and land surface air temperature (LSAT) during the early 20th century (1900-1930). The authors claim that this discrepancy in LSAT and SST arises due to a cold bias in SST observations because of poorly corrected biases in the direct observations of SST.

The consequences of the presented findings are far reaching and touch many key areas of basic and applied climate science. The authors mention 3 key consequences as (1) a decrease in the 20th century warming trend, a key IPCC statistic and measure of global change, (2) a decrease in the estimated global decadal variability in SST, a key metric for quantifying the internal variability of the climate system, and (3) a pathway towards improvement agreement between observed global temperature and CMIP simulations if the SST biases are corrected. If anything, the authors are underselling the broad significance of their findings. Global Mean Surface Temperature is arguably the most important metric of global climate as a measure of global energy imbalance and therefore the forced response as well as global internal variability. Nearly every subfield of climate science, climate impacts, and climate policy uses GMST frequently in analyses and GMST is prominently used in all IPCC reports and the Paris Agreement. The authors' suggestion that the GMST needs correcting is a major, novel finding with broad impact.

The study utilizes nearly all available tools to investigate the physicality of the "cold anomaly in the early twentieth century" and I am very impressed with the methods, statistics and uncertainty quantification, variety of data sources, and robustness of the findings. From my perspective, the key finding of the study is that there is a physically im cool bias in observed SST from 1900-1930 and that this bias is not present in the LSAT record. This finding is supported with observational, climate model, and paleoclimate analyses, all of which use the best practices for including uncertainty at each step in the analysis.

The majority of my following comments are about the presentation of the findings which should be improved before publication. The main text needs substantial work to make the findings and justifications clear to a broad readership. I would suggest defining key terms and periods explicitly and making clear how lines of evidence support their main points. I was always able to determine what the authors meant in a given sentence, but often only as a result of my knowledge of historical temperature data. I will provide my edits in the line-by-line comments section, but would suggest that the authors get a detailed friendly review from a climate scientist from a different subfield to make sure the paper is broadly accessible. I suggest this as the results are broadly exciting and want to ensure the full impact of the findings. In addition, Figure 1 is critical in explaining the problem to the reader, but is very hard to interpret and should be reworked to make clear the problem addressed by the study.

In sum, the submitted study is a rigorous analysis with clear conclusions that will have broad impact. I support the publication of this article after addressing the following comments.

We thank the reviewer for the careful and positive evaluation of our study and its results. We also appreciate the critique. We provide point-by-point replies to your comments below, and we have aimed specifically to change the text and presentation such that the revised paper is as accessible as possible.

Minor Comments:

(1) I suggest the authors revise the title of the manuscript to reflect that the cold anomaly is in the SSTs as well as that this cold anomaly is found in observations but believed to be erroneous due to observational biases.

We have revised the title of our manuscript. It now reads: “Early 20th century cold bias in ocean surface temperature observations”.

(2) The various HadSST4 products are critical to the conclusions of the study and not sufficiently explained in the main text. As I understand, in order of proposed accuracy from worst to best, there are (1) HadSST4-unadj (2) HadSST4 and (3) The SST fields from Cowtan-HybridSST which has coastal corrections to SST from relevant LSAT measurements. I infer various points that are not made clear and are necessary for interpreting the presented results:

HadSST4-unadj does not include the bias corrections to observations that are included in operational HadSST4

HadSST4 is in fact operational HadSST4 following Kennedy et al. (2019)

Cowtan-HybridSST is a processed version of operational HadSST3 with corrections to SST following Cowtan et al. (2018).

We expect the accuracy of products to be ranked (1) to (3) from worst to best as we expect the correction of known biases in SSTs to improve the estimation of global mean temperature.

I would suggest that the authors outline these products early in the main text. I would also suggest naming Cowtan-HybridSST differently to make clear that it is most closely comparable to HadSST4.

Thank you for this suggestion, this is very useful. We have added a sentence early on in the manuscript which introduces the datasets that we focus on (L85-89): “Furthermore, we also reconstruct GMST from the unadjusted HadSST4 fields (‘HadSST4-unadj’; raw gridded data prior to any applied corrections; same coverage as HadSST4); a hybrid SST dataset with corrections inferred from coastal weather stations (Cowtan et al. 2018) (‘CoastalHybridSST’), and night-time marine air temperatures (Cornes et al., 2020) (‘ClassNMAT’) using the same methodology.”. Also, we have revised the paragraph that follows such that we focus now on those SST products.

We have also carefully streamlined the terminology across the manuscript and figures, following suggestions from both reviewers, such that we hope that the manuscript is more easily readable and understandable now.

(3) Related to comment (2), I have recently reviewed work by some of the authors on the manuscript (Chan et al. 202X) on a novel, bias-corrected SST product. I would suggest including this data in the study in some capacity if the editor approves. I recommend some inclusion of this product, even if limited by the less developed uncertainty assessment, as the work by Chan and colleagues represents the state of the art in SST bias corrections and I expect any analyses using this product will further support the presented conclusions.

Thank you for pointing this out. We (=all authors) are aware of that upcoming paper, and we agree that it is clearly relevant, and the new dataset presents an exciting opportunity and way forward for the entire community. We have enquired with the authors of Chan and colleagues; but because the work is -as of now- still unpublished there may still be small changes to be expected before this new dataset is fully usable. Therefore we would prefer to keep focussing on the HybridCoastalSST dataset in this paper (which follows a similar logic as the future Chan et al., dataset; and is currently published), rather than showing a premature unpublished estimate which may still change in the future.

We strongly agree with the reviewer that our analysis will support the approach taken by Chan and colleagues, namely homogenising the SST record with coastal data.

(4) Figure 1 is currently very difficult to read and should be redesigned. The colors do not allow distinction between HadSST4-unadj and Cowtan-HybridSST or CRUTEMP5 and BEST-Land without context. In my view, the most powerful comparisons here are CRUTEMP5 with the three SST products discussed in comment (2). Thus, I would suggest that the authors simplify panels a-f in some manner by either removing or deemphasizing other series. The other series do provide useful context and support for the findings, but currently serve more as a distraction.

Another potential way to make the figure more impactful is to provide a zoom in to the time period of interest. One panel could show the entire record, but the remainder could focus on 1870-1950, for example. This could help highlight the time period of interest and make the figures more interpretable. If you do not do a zoom in, it could be helpful to make key time periods that are referred to in the text with faint horizontal lines to align the reader to the interesting time periods.

Finally, make clear that the implied SST adjustments are with respect to HadSST4-unadj in the subfigure title.

Thanks for these suggestions, again we think these are useful revisions to make the paper more easily understandable. We have changed the following:

- As suggested, we have omitted the lines of the alternative SST products (ERSSTv5 and COBE-SST2), and the Berkeley Earth land air temperature datasets. These lines are now shown in Extended Data Figure 5; and thus in Fig. 1 we emphasize the key datasets as outlined above. We have also changed the lines to bold and reworked the colours, such that the five lines in Fig. 5 should be visually well distinguishable.
- We have added the 1900-1930 period as grey shading (as well as some other lines that should improve visual guidance in the figure).
- Extended Data Figure 5 now provides a “zoom-in” into the early record, as well as showing all datasets including those that are omitted now from Fig. 1.
- We have improved the figure labels and caption. In particular, it should now come out more clearly that panel (g) shows the adjustments with reference to HadSST4-unadj.

(5) There are various time periods discussed in this study, but they are not always clearly defined in the text. I would suggest always writing about a time period by using the years of the period and sticking to as few of these periods as possible to improve continuity. These periods could be denoted visually in key figures as well to better connect the text and the figures as suggested in comment (4) for Figure 1.

We have improved the link between text and figures in highlighting the different periods. Figure 1 and Figure 2 now show a light grey shading on the 1900-1930 period, and we have added vertical lines in 25-year intervals to facilitate figure interpretation.

Likewise, we have revised the writing, such that when other periods are used this is clearly highlighted (the two other periods used are the 1901-20 vs 1870-90 average temperature change in Fig. 3 and Fig. 4; and the constraints on multidecadal warming trends in Fig. 5 looking at 1871-1910 and 1901-40 trends).

(6) The section on “Marine data sources during the ocean cold anomaly” at L156 felt out of place and longer than necessary as it doesn’t present any findings. I would suggest summarizing this section in a sentence or two in the introduction and/or conclusion. The reclaimed words would be useful in more completely spelling out the presented findings.

We have carefully revised, sharpened and slightly shortened this section, and we suggest to leave it in place because we do think it provides useful context to a broad readership about how raw SSTs are corrected (see comment in that direction by Reviewer #1). We also think that this section is useful because it is independent evidence for biases in SST data that doesn’t depend on models or other data sources (more analysis on this independent line of evidence for SST biases in this period is available in Supplementary Information).

We have also revised the Supplementary analysis, and moved to figures from Supplement to Extended Data (Extended Data Figure 7-8), in order to better integrate that section and to make the case for potential, distinct country-based SST biases more clearly.

(7) The language used to describe the ridge regression model is awkward at times and not consistent throughout the text and methods. I would suggest using either the word “predict” or “estimate” to signal when ridge regression is being used to predict a time series from a field. This will clean up the main text as well as dramatically improve the readability of the methods. Also, make clear what language you are going to use for your target time series or predictand. At times, the word “source” is used which is non-standard and confusing.

Thanks for pointing this out. We have revised the text that describes the ridge regression model: We now consistently use the word “predict” when it is about the ridge regression predictions of a (global-scale) temperature target metric based on a spatial field. The word “estimate” is used in contrast when we talk about the gridded bias and uncertainty “estimates” from CRUTEM5 and HadSST4, and when we talk about the estimation of ridge regression coefficients. We checked that we talk about “target metrics” throughout the text, and we avoid the word “predictand”. We have also revised the text to avoid the word “source”, which is now used only in the context of different SST data sources.

Line-by-line Comments

L25: Make clear here that you show that the SST cold anomaly is unforced and that IV can't explain. This is a super exciting finding of the study that could be better highlighted here.

We have revised the sentence to read: "The ocean cold anomaly is unforced, and internal variability in climate models cannot explain the observed land-ocean discrepancy." (so the two half-sentences are better connected than in the previous version)

L40: It would be helpful to orient the reader that there are three difficulties in assessing the quality and consistency of SSTs/LSATs before going into detail: (1) Observational biases (2) Interpolation/Extrapolation (3) The complexity of the climate system.

Thanks, good idea, we've implemented this additional guidance.

L49: This sentence leads me to ask why the LSAT record is judged to be more reliable. Either provide a short explanation or make this claim stronger if supported by previous studies.

We have clarified this sentence (L56-57): "Yet, at large spatial scales the LSAT record is considered more reliable because the bias adjustments are smaller and less systematic than for SSTs (Jones et al., 2016)." Jones (2016) argue that the uncertainties are likely to scale with the size of the adjustment and SST adjustments are larger, especially when analysed at large spatial scales. To summarize briefly:

- Section 3 (Issues to consider in series adjustment and error assessment): "As will be shown in this paper, the greatest uncertainty is in the marine data before World War 2"
- Section 6: "The recent study by Karl et al. (2015) illustrates that SST adjustments are by far the largest factor impacting hemispheric and global temperature measurements"
- Section 4.1 (SST measurements) "The importance of these measurement technique biases is evident from the average size of the adjustment across the world's oceans—canvas bucket measurements need to be raised by about 0.4°C between 1900 and 1941 compared to engine intakes. [...] Temperatures measured in wooden buckets before the 1890s must also be raised relative to engine intake measurements, but by smaller amounts than for canvas buckets [...] These uncertainties are dependent on the size of the adjustments, so are larger for the canvas as opposed to wooden buckets. Thus, even though coverage is sparser in the late 19th century, the uncertainties are larger between 1910 than 1940 than those from the earlier sparser coverage."

L51-53: The sentence beginning with "research focused on filling these data gaps" felt out of place. I'm not sure the goal of this sentence here. Furthermore, most operational products do the infilling separately for LSAT and SST so I would somewhat disagree with the second half of the sentence even if new methods have focused on the full global field.

Thanks for pointing this out. We have removed this sentence.

L66-67: Make clear that this is the "statistical model" as you were just discussing CMIP6 climate models.

done

L98 “extended archive” of what?

This half-sentence has been removed in the revision of the full paragraph related to your comment above.

L109: I would suggest a paragraph break before “Next, we focus on attribution”. Also, I would suggest clearly spelling out that you have shown in Figures 1d-f that the forced response and interannual variability cannot explain the difference between ocean-derived estimates, leaving the decadal IV as the discrepancy. You do say this, but taking a few more lines to really make this clear will make the next analysis more impactful to the reader.

We have introduced a paragraph break as suggested. We have revised the paragraph on attribution and internal variability to make the findings more explicit, ending this paragraph with a mini summary (L151-153): “Hence, neither a response to forcing nor internal variability can explain the ocean cold anomaly, unless all state-of-the-art climate models would miss a key, unknown process leading to multi-decadal decoupling of the ocean and land temperature record.”

L116: replicate -> replicates

We have replaced “replicate” by “captures”, because it better captures what we meant.

L118: I suggest adding “[SST/LSAT] reconstructions during the ocean...”

We added: “... observed difference between ocean- and land-based reconstructions during the ocean cold anomaly...” (now L144-145)

L121: Should be Fig2d and Fig 2e?

Yes, fixed.

L124: It could be helpful here to include a sentence explicitly stating that your CMIP analysis suggests that the cold anomaly cannot be explained by either the CMIP6 forced response or internal variability to really drive this point home. You have shown this here, but don’t directly say it.

Thanks for the suggestion! We included that sentence: (L151-153): “Hence, neither a response to forcing nor internal variability can explain the ocean cold anomaly, unless all state-of-the-art climate models would miss a key, unknown process leading to multi-decadal decoupling of the ocean and land temperature record.”

L143: I suggest removing “upper range” unless supported by values. To eye, they appear to simply “align”

Yes, we reworded the sentence.

L144-145: It is not clear what is meant by the second half of the sentence, “highlighting a discrepancy...”

Half-sentence is removed.

L183: Make clear that you are talking about the 1900-1930 or “early 20th century” cold anomaly again here.

done

L184: Briefly Introduce the ETCW here or earlier in the text as it is not a ubiquitously known problem.

the ETCW is introduced now with the approximate period of warming (L222-223): “the early twentieth century warming (ETCW) from approximately 1900-1940 has been studied widely (Hegerl et al., 2018), but the contributing causal factors remain relatively poorly understood.”

L187-190: It could be clarifying/helpful to edit or append this sentence to put this finding in the context of external forcing vs. internal variability.

We have revised and expanded this paragraph, which we think reflects your comment (L223-233): “Attribution studies of the observed early warming suggest that about half of this warming is due to external anthropogenic and natural factors (Hegerl et al., 2018), implying a significant role for internal multi-decadal variability (Delworth et al., 2000; Hegerl et al., 2018, Egorova et al., 2018). Yet, studies that resolve land and ocean temperatures separately (Stott et al., 2000; Hegerl et al., 2018) show that a much larger fraction of early land temperature changes can be attributed to external anthropogenic and natural forcing, leaving only a smaller role to multi-decadal variability. Observed SSTs are cooler than models' SSTs around the turn-of-the-twentieth-century, and exhibit fast warming thereafter (Stott et al., 2000; Hegerl et al., 2018), thus implying the large multi-decadal internal variability stems primarily from the SST record. In summary, if the ocean cold anomaly would be considered as real, one would conclude that the models underestimate the decoupling between ocean and land warming trends (Fig. 5a), for reasons still unknown.”

L202: Should this be “global temperature [interannual] variability” as there is not agreement on decadal scales as discussed?

We specified (L250-251): “Our GMST reconstructions from land and SST data yield a consistent picture of global interannual temperature variability and long-term changes during the instrumental period, yet they [...]”

L216: It is not clear what “constraints” refers to.

We realize this has been insufficiently explained in the submitted manuscript. We added a comprehensive explanation on how the emergent constraint technique (Cox et al., 2013) is used (P40-41, “Derivation of observational constraints on historical ocean warming in Fig. 5”), and we have carefully revised Fig. 5. Please see our reply to a similar comment from Reviewer #1 (her/his Comment 7).

Last sentence: The last sentence should be much stronger. The presented study has provided multiple lines of robust evidence that the current SST record is incorrect in the early 20th century and the closing sentence, and conclusion, should reflect this.

We have reformulated the last paragraph overall (including some changes made in response to Reviewer #1), and we have revised the last sentence. It reads now (L270-274): “Yet, the presented constraints based on coastal land temperatures (Cowtan et al., 2018; Chan et al., 2023), statistical analysis and paleoclimate data imply that agreement between models and reality in the early twentieth century is higher than current observational datasets suggest, and that the role of multi-decadal temperature variability is smaller than previously thought. “

Methods p21: It appears the three components are treated as statistically independent. If this is the case, make this clear when introducing them.

Yes, they are assumed to be statistically independent. Fixed.

Methods p22: How are the uncorrelated uncertainties estimated with gridded error fields?

We changed the description to say “uncorrelated uncertainties are available in the form of gridded error fields”.

Methods p27: I suggest adding a paragraph break before “In addition”

Done

Methods p27: The description of the additional reconstructions is not clear. I would suggest editing this entire section following my suggestion in minor comment (7).

We have revised this paragraph (p. 34 bottom, p. 35). We hope it is clear now.

Methods p31: Define “Turn of the 20th century” in years in the title.

done

Figure 4: It could be helpful to add a horizontal line at 0 to provide a reference for readers to compare distributions in both panels a and b.

We have added the horizontal lines in panels a and b.

Additional references (which are not part of the manuscript)

Allen, R.J. and Sherwood, S.C., 2008. Warming maximum in the tropical upper troposphere deduced from thermal winds. *Nature Geoscience*, 1(6), pp.399-403.

Fu, Q., Johanson, C.M., Warren, S.G. and Seidel, D.J., 2004. Contribution of stratospheric cooling to satellite-inferred tropospheric temperature trends. *Nature*, 429(6987), pp.55-58.

Karl, T.R., 2005, January. Temperature trends in the lower atmosphere: understanding and reconciling differences. In *16th Conference on Climate Variability and Change*.

Meehl, G.A., Washington, W.M., Ammann, C.M., Arblaster, J.M., Wigley, T.M.L. and Tebaldi, C., 2004. Combinations of natural and anthropogenic forcings in twentieth-century climate. *Journal of Climate*, 17(19), pp.3721-3727.

Lyman, J.M., Willis, J.K. and Johnson, G.C., 2006. Recent cooling of the upper ocean. *Geophysical Research Letters*, 33(18).

Sherwood, S.C., Lanzante, J.R. and Meyer, C.L., 2005. Radiosonde daytime biases and late-20th century warming. *Science*, 309(5740), pp.1556-1559.

Reviewer Reports on the First Revision:

Referee #1 (Remarks to the Author):

I won't repeat my comments about the key results, originality, significance etc. which I already gave in my first review, other than to confirm that they still apply to the revised manuscript and therefore I still recommend the manuscript be published in Nature.

Here I will just focus on the revision and response to reviewers.

I believe the responses to both reviews are very good. The responses are clear, the comments have been carefully considered (including some extra analyses to demonstrate a particular point) and either changes have been made or compelling reasons for not making changes have been given. I agree with the rebuttal points and I have looked at the changes that have been made to the manuscript and I think they are all appropriate. I recommend that the manuscript be published in Nature.

I have some very minor comments to make. There is no need for me to see the manuscript again, it is up to the authors and editor if any changes are needed for these very minor comments, and I do not need to check them.

(1) In the merged PDF manuscript file, the legend inset into panel (a) of Fig 5 has gone wrong, with symbols mis-aligned or missing from legend items.

(2) In response to my original review comment 10, the authors explain that they have extracted a CRUTEM5 ensemble from the HadCRUT5 and HadSST4 ensembles. The rebuttal response explains how this was done and this looks like a correct method, but no change to the manuscript was made. Perhaps add one sentence to the methods section to tell the reader that you did this extraction? Perhaps the addition in italics here would work?

"A 200-member ensemble of potential realizations of known, temporally correlated uncertainties in near-surface air temperature has been produced as part of the HadCRUT5 Noninfilled Data Set13 (i.e., $\epsilon_{b, \text{Land}(s, t)}$). The corresponding ensemble of CRUTEM5 LSAT anomalies has been extracted from this, using the HadSST4 ensemble to unblend SST and LSAT anomalies in coastal grid cells."

(3) In the response to my original review comment 10, the authors state "Regarding urbanization and exposure biases: These biases do have a spatial and temporal structure and are encoded in the ensemble realizations of the CRUTEM5 ensemble according to Morice et al (2021)." Yet modified manuscript still says this "For LSAT, there is no spatially correlated error term (i.e., $\epsilon_{p, \text{Land}(s, t)} = 0$)." This seems inconsistent with the above response.

Referee #1 (Remarks on code availability):

I have checked that the GitHub repo is present and a readme file is present. I have not attempted to run the code.

Referee #2 (Remarks to the Author):

Dear authors,

Thank you for addressing my comments so carefully. In particular, Figure 1 is now much clearer and tells the story of the study very well.

I recommend acceptance with no additional edits.

Referee #2 (Remarks on code availability):

The code is very well documented through comments and a detailed README and easy to run.

Author Rebuttals to First Revision:

We would like to thank the reviewers and the editor -again- for the positive evaluation of our manuscript, and we reply to the comments below.

Referees' comments:

Referee #1 (Remarks to the Author):

I won't repeat my comments about the key results, originality, significance etc. which I already gave in my first review, other than to confirm that they still apply to the revised manuscript and therefore I still recommend the manuscript be published in Nature.

Here I will just focus on the revision and response to reviewers.

I believe the responses to both reviews are very good. The responses are clear, the comments have been carefully considered (including some extra analyses to demonstrate a particular point) and either changes have been made or compelling reasons for not making changes have been given. I agree with the rebuttal points and I have looked at the changes that have been made to the manuscript and I think they are all appropriate. I recommend that the manuscript be published in Nature.

I have some very minor comments to make. There is no need for me to see the manuscript again, it is up to the authors and editor if any changes are needed for these very minor comments, and I do not need to check them.

(1) In the merged PDF manuscript file, the legend inset into panel (a) of Fig 5 has gone wrong, with symbols mis-aligned or missing from legend items.

Thank you for checking carefully, we have indeed missed this. The issue has been corrected.

(2) In response to my original review comment 10, the authors explain that they have extracted a CRUTEM5 ensemble from the HadCRUT5 and HadSST4 ensembles. The rebuttal response explains how this was done and this looks like a correct method, but no change to the manuscript was made. Perhaps add one sentence to the methods section to tell the reader that you did this extraction? Perhaps the addition in italics here would work?

“A 200-member ensemble of potential realizations of known, temporally correlated uncertainties in near-surface air temperature has been produced as part of the HadCRUT5 Noninfilled Data Set13 (i.e., $\epsilon_{b, \text{Land}}(s, t)$). The corresponding ensemble of CRUTEM5 LSAT anomalies has been extracted from this, using the HadSST4 ensemble to unblend SST and LSAT anomalies in coastal grid cells.”

Thank you for catching this, indeed we have explained the procedure in the previous response, but had left the rather unspecific explanation in the manuscript. We adopt your suggestion to be specific about how the unblending was done for coastal grid cells.

(3) In the response to my original review comment 10, the authors state "Regarding urbanization and exposure biases: These biases do have a spatial and temporal structure and are encoded in the ensemble realizations of the CRUTEM5 ensemble according to Morice et al (2021)." Yet modified manuscript still says this “For LSAT, there is no spatially correlated error term (i.e., $\epsilon_{p, \text{Land}}(s, t) = 0$.” This seems inconsistent with the above response.

We believe this is a potential misunderstanding, thanks for highlighting: In the CRUTEM5 ensemble of bias realizations, spatially correlated errors exists (of course), and are part of the bias realizations ($\epsilon_{b, \text{Land}}(s, t)$), as indicated in a sentence a few lines earlier: “*In CRUTEM5, uncertainties are encoded following the method given in ref. 52. A 200-member ensemble of potential realizations of known, temporally and spatially correlated uncertainties in near-surface air temperature has been produced as part of the HadCRUT5 Noninfilled Data*

Set13 (i.e., $\epsilon_{b, \text{Land}}(s, t)$). The corresponding ensemble of CRUTEM5 LSAT anomalies has been extracted from this, using the HadSST4 ensemble to unblend SST and LSAT anomalies in coastal grid cells. The ensemble realization of biases encompasses uncertainties such as station-based homogenization errors and uncertainty in climatological normals, as well as regional urbanization errors and nonstandard sensor enclosures (full description provided in refs.13,24,52).”

We have included “spatially correlated uncertainties” in the sentence quoted above to be explicit. To avoid any potential misunderstandings, we have removed the sentence that mentions the term $\epsilon_{p, \text{Land}}(s, t)$ (with the sentence about $\epsilon_{p, \text{Land}}(s, t)$ we originally wanted to indicate just that the term ϵ_p , which exists for the ocean ensemble, does not exist for the land because the spatially and temporally correlated error terms are part of the bias realizations).

Referee #1 (Remarks on code availability):

I have checked that the GitHub repo is present and a readme file is present. I have not attempted to run the code.

Referee #2 (Remarks to the Author):

Dear authors,

Thank you for addressing my comments so carefully. In particular, Figure 1 is now much clearer and tells the story of the study very well.

Thank you again for the positive and constructive review comments.

I recommend acceptance with no additional edits.

Referee #2 (Remarks on code availability):

The code is very well documented through comments and a detailed README and easy to run.